# Non-Markovian relaxation spectroscopy of fluxonium qubits

Ze-Tong Zhuang[1], Dario Rosenstock[1,4], Bao-Jie Liu [1], Aaron Somoroff[2,5], Vladimir E. Manucharyan [3] & Chen Wang [1] ✉

Recent studies have shown that parasitic two-level systems (TLS) in superconducting qubits, which are a leading source of decoherence, can have relaxation times longer than the qubits themselves. However, the standard techniques used to characterize qubit relaxation is only valid for measuring $T_1$ under the Born-Markov approximation and could mask environmental memory effects in practice. Here, we introduce two-timescale relaxometry, a technique to probe the qubit and environment relaxation simultaneously and efficiently. We apply it to high-coherence fluxonium qubits over a frequency range of 0.1-0.4 GHz, and reveal a discrete spectrum of TLS with millisecond lifetimes. Our analysis of the spectrum is consistent with a random distribution of TLS in the aluminum oxide tunnel barrier of the Josephson junction chain of the fluxonium, with a spectral and volumetric density and average electric dipole similar to previous TLS studies at much higher frequencies. Our study suggests that investigating and mitigating TLS in the junction chain is crucial to the development of various types of noise-protected qubits in circuit QED.

Qubit relaxation in most quantum technology platforms is commonly modeled as arising from interactions with a Markovian environment that has no memory of its prior interaction with the qubit. The Born-Markov approximation[1] breaks down when there are hidden environmental degrees of freedom with long relaxation times, which can lead to unexpected spatio-temporal correlations in the system dynamics and ultimately pose challenges to the implementation of quantum algorithms and error correction protocols[2–4]. While some of the non-Markovian effects in qubit dephasing have been widely recognized (such as the ubiquitous 1/f noise[5]), there has been relatively limited attention to non-Markovian effects in common practices of characterizing qubit relaxation dynamics. For superconducting qubits, both spurious two-level systems (TLS)[6,7] and quasiparticle excitations[8] in the environment bath have occasionally shown long-time memory of their past energy exchange with the qubit[9–14]. As a result, qubit relaxation becomes apparently non-Markovian as it shows a dependence on its past operation history. These prior studies typically employ ad hoc pulse sequences to initialize the bath to different states

before proceeding with a standard (pump-delay-probe) qubit $T_1$ measurement that may inform joint relaxation of the qubit and the bath. Given that the most coherent superconducting qubits today are $T_1$-limited[15–17], systematically assessing the role of potential non-Markovian relaxation dynamics is essential for establishing a more complete picture of the frontier of qubit coherence.

In particular, the superconducting fluxonium qubits[18] present an intriguing case where non-Markovian effects may play a prominent role in their coherence. The fluxonium recently emerged as a promising building block for quantum processors, having demonstrated among the best coherence times (≥1 ms[17,19]) and two-qubit gate fidelities (99.9%[20–22]) in superconducting circuits. However, its more sophisticated device construction (requiring a chain of Josephson junctions[18,23] or long strips of high-kinetic-inductance materials[12,24,25]) makes it more likely to host a large number of TLS; the relatively low qubit frequency (<1 GHz) implies that the resonant TLS may potentially have slow dynamics (e.g., $\propto \omega^3$ for phonon-mediated relaxation in the standard tunneling model in 3D[6]). Indeed, a recent study of a

[1]Department of Physics, University of Massachusetts-Amherst, Amherst, MA, USA. [2]Department of Physics, University of Maryland, College Park, MD, USA. [3]Institute of Physics, Ecole Polytechnique Federale de Lausanne, Lausanne, Switzerland. [4]Present address: Google Quantum AI, Santa Barbara, CA, USA. [5]Present address: SEEQC, Inc., Elmsford, NY, USA. ✉e-mail: wangc@umass.edu

fluxonium qubit made of granular aluminum revealed a TLS bath with millisecond relaxation times compared to a relatively short qubit lifetime on the order of 10 $\mu s$[12]. Ref. 12 also introduced a powerful technique dubbed the "quantum Szilard engine" that uses fast feedback control of the qubit to manipulate the bath while using the feedback-control records to characterize the dynamics of the joint system. However, this technique relies on high-fidelity readout with near-perfect quantum non-demolition (QND) properties, which requires system optimization and may not be available in typical settings of $T_1$ spectroscopy studies when the measurement quality varies widely across frequencies and across devices.

In this work, we introduce a more adaptable measurement and analysis framework designed to probe the qubit and environment relaxation simultaneously and efficiently, which we refer to as two-timescale relaxometry. The key principle is to consistently extract information on the qubit transition (decay or excitation) rates from $T_1$-like measurements on short intervals while attempting to polarize the bath on long timescales. This method requires faster reset of the qubit than bath relaxation but does not require QND readout. It can tolerate substantial infidelities in qubit readout and reset assuming they can be calibrated, and overall yields similar data throughput as conventional $T_1$ measurements.

We apply two-timescale relaxometry in a spectroscopic study of fluxonium qubits in the frequency range of 130-400 MHz. We find that the qubit relaxation dynamics is dominated by a discrete spectrum of TLS with millisecond $T_1$ but sub-microsecond $T_2$ times. Although our study does not distinguish the microscopic origins of these TLS, finite-element modeling of the fluxonium mode's electric-field distribution shows that the relaxation spectrum is well explained by a random distribution of prominent TLS in the aluminum oxide tunnel barrier of the Josephson junction chain, together with a much weaker continuum of surface dielectric loss (on the order of ms⁻¹). The observed number and coupling strength of these TLS correspond to an average area density of 0.4 GHz⁻¹μm⁻² and average effective dipole of 6 Debye for TLS residing in the junction chain. Our study, therefore, identifies a set of TLS in a frequency range that can influence typical quantum circuits as hidden slow-switching "thermal fluctuators", with densities and dipoles similar to those more widely investigated TLS at multi-GHz frequencies[7]. We conclude by discussing the implications of these findings for improving coherence times in fluxoniums and other protected superconducting qubits.

## Results

### Non-Markovian dynamics in fluxonium qubits

Our main study is carried out on a high-coherence fluxonium qubit (Fig. 1 (a)) inside a 3D copper cavity ($\kappa/2\pi$ = 8.7 MHz and $\chi/2\pi$ = 0.41 MHz), which demonstrated coherence times that exceed 1 ms at the half-flux point in a previous study[17]. The fluxonium has $E_J/2\pi$ = 4.88 GHz, $E_C/2\pi$ = 1.09 GHz, $E_L/2\pi$ = 0.56 GHz, and half-flux frequency $\omega_{01}/2\pi$ = 198 MHz. The best coherence times observed in this study are slightly lower than in Ref. 17 due to stochastic presence of TLS near the half flux point and less rigorous filtering and shielding in a different cryogenic setup, but nonetheless reaches $T_2^*$ = 320 $\mu s$ and $T_2$ = 600 $\mu s$ (with Hahn echo) at half flux when not apparently impacted by near-resonant TLS. Due to the large $\kappa/\chi$ ratio and the absence of a quantum-limited amplifier, qubit readout is carried out with a 15 $\mu s$ square pulse with approximately 100 average photons, which is often destructive (i.e., non-QND) to the qubit state[26]. We also measured another fluxonium device in a planar package with half-flux $T_{2E}$ times in the range of 70 to 90 $\mu s$ with single-shot QND readout. Details of both devices are included in Methods.

The large volume of aluminum oxide in a fluxonium's junction chain potentially provides a rich soil for TLS with significant coupling to the qubit (Fig. 1b). While this interaction is not expected to reach the strong coupling regime, $g_k > \Gamma_{2,k}$, (where $g_k$ is the flip-flop interaction

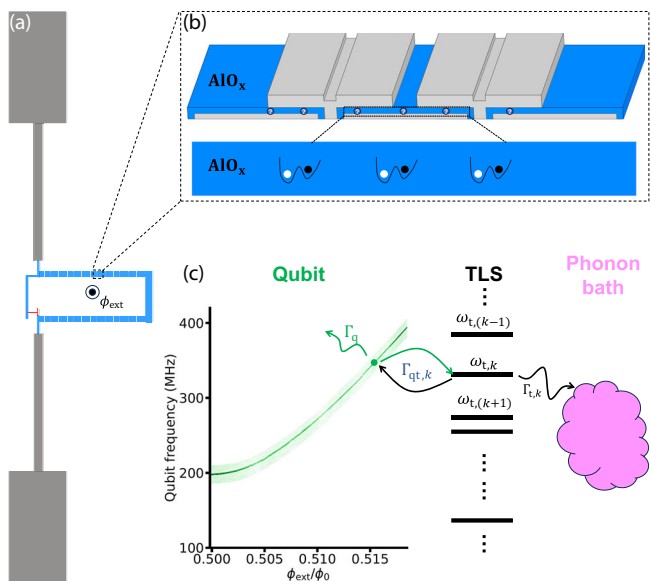

**Fig. 1 | Two level systems (TLS) in the fluxonium junction chain. a** Schematic of a superconducting fluxonium qubit consisting of a small Al/AlO$_x$ phase-flip junction (red), a long Al/AlO$_x$ junction chain (166 junctions in our main device) serving as a large inductor (blue), and a pair of antenna capacitor pads (grey). **b** Schematic of TLS residing in the oxide layer of the fluxonium's junction chain. **c** Measured spectrum of the fluxonium $|0\rangle - |1\rangle$ transition in the frequency range of this study and a cartoon depiction of the qubit-TLS relaxation dynamics. When the fluxonium is operated at a certain frequency ($\omega_q$) (tunable by external flux $\phi_{ext}$), its energy relaxation rate can be greatly affected by one or more TLS in the junction chain, with incoherent qubit-TLS energy exchange rate $\Gamma_{qt,k}$. Each TLS has its own intrinsic decay rate $\Gamma_{t,k}$ to its own bath. Other environment channels like quasiparticles and radiation modes, as well as the continuum TLS bath, which are weakly coupled to the fluxonium qubit at that frequency, contribute to a background rate of qubit decay $\Gamma_q$.

between the qubit and the $k$-th TLS, and $\Gamma_{2,k}$ is their combined decoherence rates,) it can lead to an incoherent energy exchange rate $\Gamma_{qt,k} = \frac{2g_k^2\Gamma_{2,k}}{\Gamma_{2,k}^2 + \Delta_k^2}$ that dominates over other relaxation timescales of the system. Here, $\Delta_k$ is the frequency detuning between the qubit and the TLS. In this incoherent limit, qubit-TLS relaxation dynamics can be described by the Solomon equations[27,28] (Fig. 1c):

$$\frac{dZ}{dt} = -\Gamma_q(Z - Z^{eq}) - \sum_k \Gamma_{qt,k}(Z - p_k), \quad (1)$$

$$\frac{dp_k}{dt} = -\Gamma_{t,k}(p_k - p_k^{eq}) - \Gamma_{qt,k}(p_k - Z), \quad (2)$$

where $Z = \langle\sigma_z\rangle$ represents the qubit polarization in energy eigenbasis, with $Z^{eq} = \langle\sigma_z\rangle^{eq}$ its equilibrium value at long times. $p_k$ is the polarization of the $k$-th TLS, and $p_k^{eq}$ its corresponding equilibrium polarization. $\Gamma_q$ is the background decay rate of the qubit, and $\Gamma_{t,k}$ are the decay rates of the TLS to their own relaxation channels, e.g. the phonon bath. A general solution to these equations leads to non-Markovian qubit relaxation dynamics in the sense that it violates the classical definition of Markovian processes[29], although certain quantum information theoretical definitions of Markovianity may differ (e.g., ref. 30). This dynamics could manifest as multiple exponential relaxation timescales in experiments if the entire coupled system can be reproducibly initialized.

To illustrate a prominent case of non-Markovian relaxation dynamics, we show qubit $T_1$ relaxation measurements following different bath-preparation sequences (similar to Refs. 9,10), as shown in

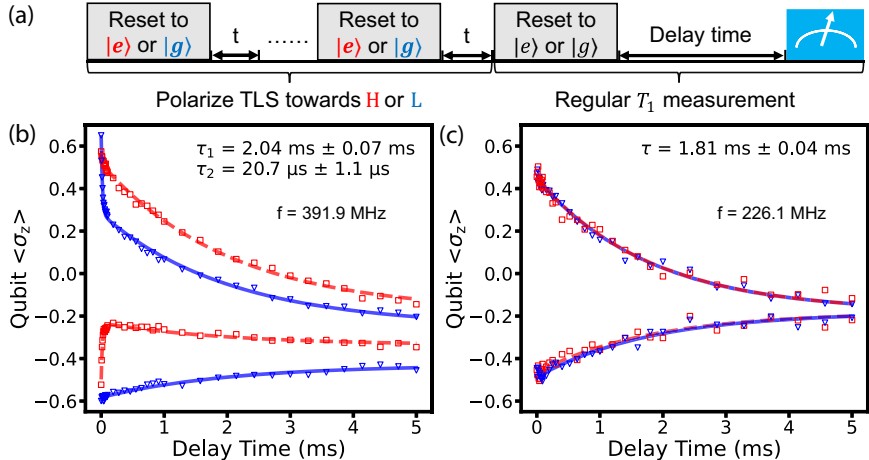

**Fig. 2 | Regular qubit-$T_1$ measurements following bath-preparation sequences.** **a** Pulse sequence diagram. Before a $T_1$ measurement sequence, we polarize the bath (i.e., the TLS state) to a high-energy state or a low-energy state by repeatedly initializing the qubit to $|e\rangle$ or $|g\rangle$ and allowing for a qubit-TLS energy exchange time $t$. In this example, we use $t = 25\,\mu$s and reset the qubit 7 times. After the TLS state is prepared, regular $T_1$ measurements are performed with the qubit initialized in $|e\rangle$ or $|g\rangle$ states. **b** Example qubit relaxation data, $\langle\sigma_z\rangle$ as a function of delay time $t$, acquired at one specific flux point. Red points correspond to TLS initialized in the high-energy state, and the blue points correspond to the low-energy state. All data is fit with the sum of two exponential decay functions using the same two time scales. One is the qubit-TLS interaction rate, i.e., $2\Gamma_{\rm qt}$, and the other corresponds to the qubit and TLS decaying together, i.e. $\frac{\Gamma_q + \Gamma_t}{2}$[28]. **c** Example qubit relaxation data acquired at a different flux point. As red and blue traces overlap, the bath is largely not polarizable at this qubit frequency.

Fig. 2a. Here, the qubit reset is realized with a strategy of applying a 35 $\mu$s intentionally non-QND readout pulse to the cavity[31], giving reset fidelity in the range of 60% to 95% (mean reset fidelity 72%) (see Supplementary section IV for details). Such bath-preparation sequences can saturate the TLS bath towards either its ground or excited state, and the subsequent measurement of the qubit decay is carried out with the qubit prepared in either $|g\rangle$ or $|e\rangle$. When the bath contains a prominent long-lived TLS (Fig. 2b), we can use a double-exponential model to capture all four decay curves. Such a non-Markovian feature can be easily overlooked and lead to misinterpretation of the qubit relaxation dynamics if traditional $T_1$ protocols are used without attention to the state of the bath (see Supplementary Section II for examples). In comparison, at a different flux-bias point, the bath-preparation sequences may have much weaker impact on the qubit relaxation curve (Fig. 2c), indicating no significant memory effect from the bath.

Although this method illustrates the non-Markovian relaxation dynamics clearly, it is very slow: It requires a relatively long bath-preparation sequence without acquiring data, and the average interval of measurement is on the order of the slowest time scale of the system, which is milliseconds in our device and generally is not known a priori. This limitation makes it difficult to characterize the TLS environment for a wide frequency range before it reconfigures.

**Two-timescale relaxometry protocol**
To more efficiently measure non-Markovian relaxation dynamics, we present a technique that can probe qubit relaxation together with the environment relaxation on independent time scales. Our measurement protocol is shown in Fig. 3a. Within each experimental cycle, the TLS environment is first polarized towards its excited state over a long timescale ($T$) via interactions with a qubit that is repeatedly re-initialized to $|e\rangle$, and then polarized towards its ground state with the qubit repeatedly re-initialized to $|g\rangle$. Throughout this long bath-polarization cycle, we continually monitor instantaneous qubit decay and excitation rates $\Gamma_\downarrow$ and $\Gamma_\uparrow$ by inserting qubit measurements following variable qubit-TLS interaction time $t$. These short time intervals $t$ act both as the incremental interaction window to allow the qubit to polarize the bath and as a list of variable delay times to probe the instantaneous qubit relaxation rates under the existing bath condition.

If the state of the TLS bath changes slowly over time, we may group several qubit resets and readouts into a single measurement block (e.g., with four different $t$, as shown in Fig. 3(a), $0 = t_0 < t_1 < t_2 < t_3$), and the bath is treated as in a quasi-steady state within the block. Within each block, by fitting the qubit state $Z$ as a function of $t$ to an exponential decay curve, we can obtain the initial slope of the qubit relaxation $\dot{Z}_0$, i.e., the time-derivative of the qubit polarization $dZ/dt$ at $t = 0$. $\dot{Z}_0$ can be intuitively understood as a proxy for $\Gamma_\downarrow$ or $\Gamma_\uparrow$: in the limit of perfect qubit reset fidelity, we have $\dot{Z}_0 = -2\Gamma_\downarrow$ or $\dot{Z}_0 = 2\Gamma_\uparrow$ when the qubit is initialized to $|e\rangle$ or $|g\rangle$, respectively. We note that a four-point exponential fit can usually yield $\dot{Z}_0$ robustly over a broad range of qubit relaxation timescales, from $\sim t_1$ to 5-10 times of $t_3$, which motivates our choices of $t$ here. This quasi-steady-state approximation becomes inaccurate when the qubit dynamics is dominated by a single long-lived TLS. In such a case, we perform only one qubit readout per block but vary its delay time $t$ in an additional layer of sequence loops to allow accurate extraction of $\dot{Z}_0$ (Methods: Two-timescale relaxometry with cycling delay times).

In Fig. 3b, we show a typical data set of fitted $\dot{Z}_0$ as a function of the sequence block index, averaged over many bath-polarization cycles. We also associate each block with its start time within the bath-polarization cycle ($T$). This $\dot{Z}_0(T)$ data can be heuristically interpreted as the qubit $\Gamma_\downarrow$ rate starts large and saturates to a smaller value as the TLS is saturated towards the excited state in the first half cycle (red curve), and vice versa for $\Gamma_\uparrow$ rate in the second half cycle (blue curve).

When the qubit is tuned to different operating frequencies, we observe substantially different relaxation time scales. Fig. 3c shows an example of slow bath dynamics that requires more than 20 ms to saturate. Fig. 3d shows an example of fast bath dynamics. It should be noted that this fast bath dynamics is often due to strong relaxation through the qubit (which receives frequent resets) and should not be taken as a direct measure of the intrinsic TLS lifetime.

**Relaxation rate analysis**
To quantify qubit relaxation rates under the condition of imperfect qubit and bath polarizations, we independently calibrate the initial polarization of the qubit $Z_g$ and $Z_e$ under our reset protocol, and relate $\dot{Z}_0$ to qubit relaxation rates with $\dot{Z}_0 = -Z_{g/e}\Gamma_\Sigma - \Gamma_\delta$, where the subscript g/e depends on which state the qubit is initialized in, $\Gamma_\Sigma \equiv \Gamma_\uparrow + \Gamma_\downarrow$

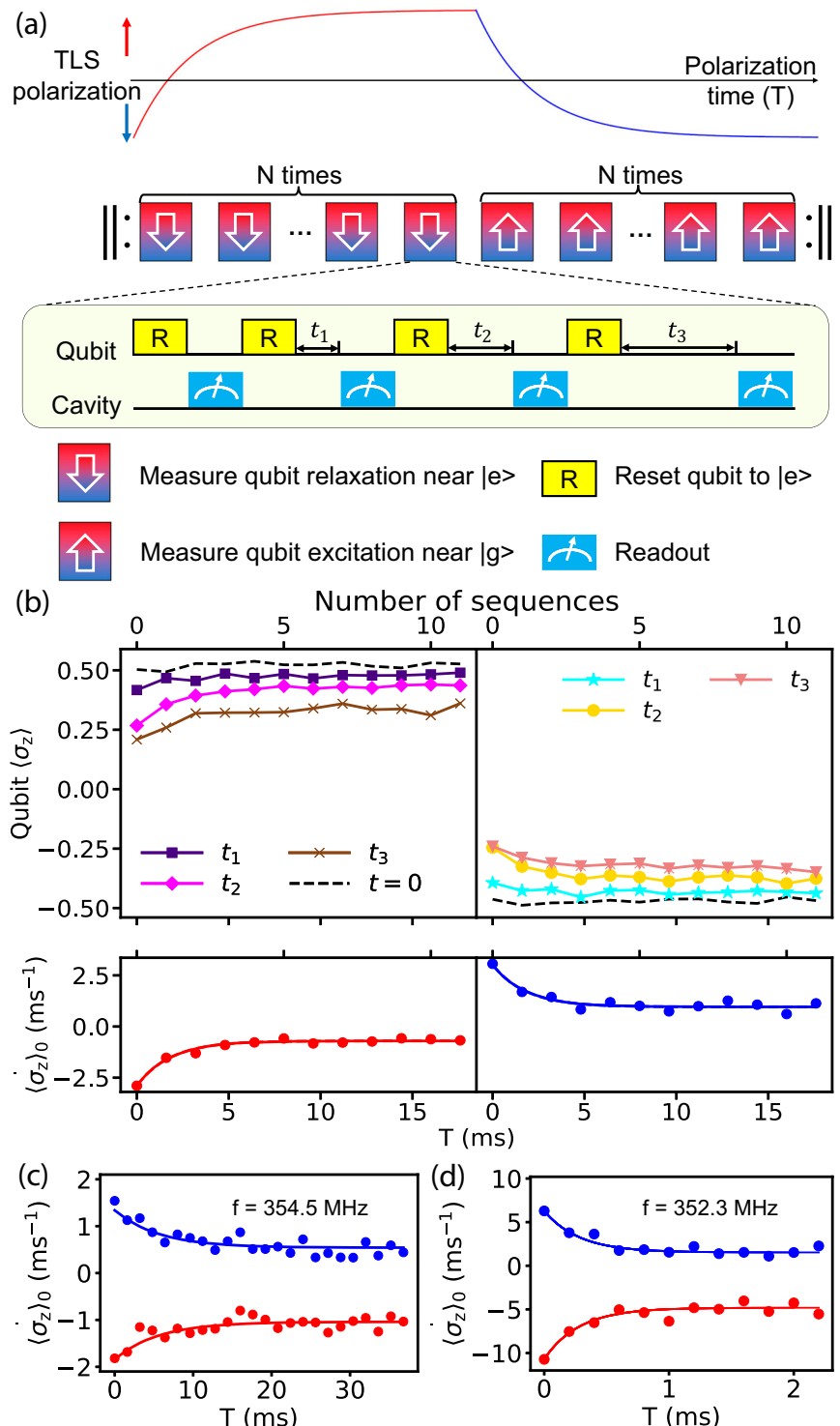

**Fig. 3 | Two-timescale relaxometry: protocol and representative data.**
**a** Schematic of the experiment. In the first half of the cycle, we aim to repeatedly extract qubit $\Gamma_\downarrow$ while we polarize TLS towards the excited state over a slow time scale of $T$, and in the second half, we do the opposite. Each cycle contains $2N$ sequence blocks, with each block containing 4 qubit resets and measurements. **b** Sample data for extracting the decay slope of qubit polarization, with $N = 12$, $t_1 = 40\,\mu s$, $t_2 = 150\,\mu s$, $t_3 = 330\,\mu s$, and block length 1.6 ms. The top panel shows the

raw data of qubit measurements calibrated to the scale of $\langle \sigma_z \rangle$. The bottom panel shows the extracted initial slope of qubit polarization, $d\langle \sigma_z \rangle / dt$ at $t = 0$, as a function of bath-polarization time $T$. A simple exponential fit (described in section C of Results) is shown with solid lines. **c** Example data showing long environmental response time, measured with an increased $N$ of 25. **d** Example data with strong qubit-TLS coupling. The block length is reduced to $200\,\mu s$ using a cyclic permutation of delay times $t_1, t_2,...$ (See Methods for more details.).

is commonly known as the inverse of the $T_1$ time, and $\Gamma_\delta \equiv \Gamma_\downarrow - \Gamma_\uparrow$ is the differential qubit relaxation rate for its down versus up transitions.

Over the long timescale of each bath-polarization cycle, the state of the bath converges between two steady states, a higher-energy state

($H$) closer to the excited, and a lower-energy state ($L$) closer to the ground. Without loss of generality, for any observable $O$ of the bath, we can write $O(T) = O_H + (O_L - O_H)f(T)$ for the first half cycle and $O(T) = O_L + (O_H - O_L)f(T)$ for the second half cycle, where $f(T)$ is a decay

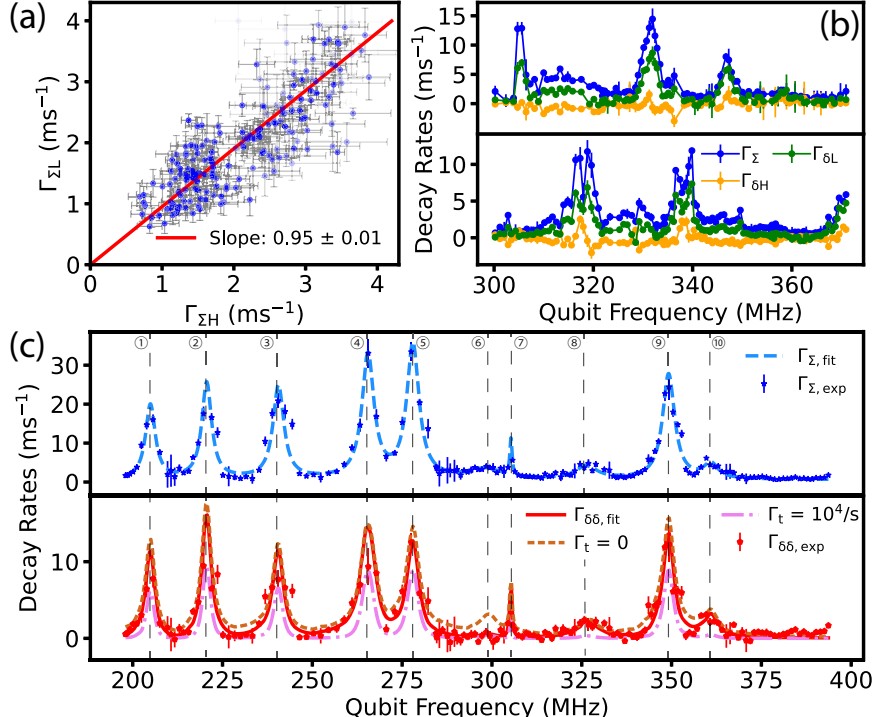

**Fig. 4 | Non-Markovian qubit relaxation spectroscopy. a** Comparison of qubit decay rate sum, $\Gamma_\Sigma$, when the TLS is polarized to high-energy vs. low-energy states, over a broad range of qubit frequencies. Here we show $\Gamma_{\Sigma L}$ vs. $\Gamma_{\Sigma H}$ when they are less than 4 ms$^{-1}$ since the uncertainties are significantly higher at large rates. Point transparency is in proportional to its uncertainty. Red line: linear fit with y-intercept fixed to 0. **b** Two example sweeps of relaxation spectroscopy two days apart, which show significant TLS reconfiguration. **c** Measured relaxation rates from a wider spectroscopy sweep. The upper panel shows a global fit of $\Gamma_\Sigma$ with 10 TLS. Each TLS is represented by a Lorentzian (Eq. (4)). The lower panel includes a global fit to $\Gamma_{\delta\delta}$ using our model (Eq. (5)) (solid line) and reference plots of the model-predicted $\Gamma_{\delta\delta}$ under two extreme conditions (dashed lines): All TLS has infinitely long ($\Gamma_t = 0/s$), and 100 $\mu s$ intrinsic lifetime ($\Gamma_t = 10^4/s$), for comparison. The fact that almost all $\Gamma_{\delta\delta}$ data we showed in the spectroscopy are higher than the $\Gamma_t = 10^4/s$ dashed line indicates TLS lifetime of more than 100 $\mu s$. Detailed fitting results are shown in Methods. All error bars represent one-standard-deviation uncertainty of fitted decay rates from the model described in the text.

function from $f(0) = 1$ to $f(\infty) = 0$. The qubit relaxation rates, $\Gamma_{\uparrow/\downarrow}$ or $\Gamma_{\Sigma/\delta}$, are proxy observables of the bath, hence we may model:

$$\Gamma_\delta(T) = \begin{cases} \Gamma_{\delta H} + (\Gamma_{\delta L} - \Gamma_{\delta H})e^{-\frac{T}{\tau_e}}, & \text{first half cycle} \\[2mm] \Gamma_{\delta L} + (\Gamma_{\delta H} - \Gamma_{\delta L})e^{-\frac{T}{\tau_e}}, & \text{second half cycle} \end{cases} \quad (3)$$

where $\Gamma_{\delta H/L}$ represents $\Gamma_\delta$ when the bath is in the H/L state. Here we have assumed a simple exponential model for the bath dynamics, $f(T) = e^{-T/\tau_e}$, adapted to the general observation and the quality of our data, but this analysis can be generalized to more sophisticated decay models and does not depend on the nature of the bath. Using Eq. (3) and its analogous equation for $\Gamma_\Sigma(T)$, we can fit $\dot{Z}_0(T)$ to obtain all the bath-dependent qubit relaxation rates $\Gamma_{\Sigma H}, \Gamma_{\Sigma L}, \Gamma_{\delta H}, \Gamma_{\delta L}$, as well as the characteristic time of bath relaxation $\tau_e$ under our measurement protocol. A more detailed derivation of modeling two-time-scale relaxometry is described in Supplementary section III.

## Non-Markovian relaxation spectroscopy

We apply the two-timescale relaxometry protocol to the 3D fluxonium device in a spectroscopy sweep from 200 to 400 MHz. We find that the measured results are mostly consistent with $\Gamma_{\Sigma L} \approx \Gamma_{\Sigma H}$ (Fig. 4a), suggesting that there is relatively little contribution to the environment memory effect from possible bosonic baths, which would have increased $\Gamma_\Sigma$ by a factor of $2\bar{n}+1$ (where $\bar{n}$ is the average occupancy number in a bosonic mode). To reduce fit uncertainties from measurement noise, in further analysis, we will constrain our model by setting $\Gamma_{\Sigma L} = \Gamma_{\Sigma H} = \Gamma_\Sigma$. In Fig. 4b we show two sample spectroscopy

sweeps that are two days apart, each showing discrete resonance peaks but at different frequencies. This is consistent with the widely-reported observations that the TLS tend to reconfigure over the timescale of hours to days[32,33], and demonstrates our ability to track TLS reconfiguration over a broad spectral range with this technique.

This experimental protocol is primarily intended as a coarse-grained spectroscopy tool that separates possible memory effects arising from slowly varying, weakly coupled environmental baths from qubit $T_1$ measurements. It can also be used to study the properties of individual TLS that dominate the qubit dynamics after incorporating shortened measurement blocks with cycling delays (see Methods). Fig. 4c shows a more elaborate example of a spectroscopy sweep that can be modeled by 10 discrete TLS resonances. Here, $\Gamma_\Sigma$ shows how strongly the environment is coupled to the qubit, while $\Gamma_{\delta\delta} \equiv \Gamma_{\delta L} - \Gamma_{\delta H}$ reflects the *polarizable part* of the environment under our protocol, i.e., the difference of the differential qubit relaxation rate comparing the highest versus lowest energy bath states accessed in the experiment. If there is only one prominent TLS, the fraction, $\frac{\Gamma_{\delta\delta}}{2(\Gamma_\Sigma - \Gamma_q)}$, reflects the TLS population difference between its high and low states. After being normalized by qubit initialization fidelity, the polarizability of the TLS is an important indication of its intrinsic lifetime.

To further extract parameters of these discrete TLS, we apply global fits of the $\Gamma_\Sigma$ and $\Gamma_{\delta\delta}$ spectra considering all prominent TLS's contribution. Unlike previous qubit $T_1$ spectroscopy studies that generally only measure $\Gamma_\Sigma$ under the Markovian assumption (except when coherent qubit-TLS oscillation is explicitly observed and modeled, e.g., ref. 34,) here we aim to model the self- and cross-relaxation rates of the qubit and its TLS environment in the frequency domain together (see

Supplementary Section III for derivations):

$$\Gamma_\Sigma = \sum_k \Gamma_{qt,k} + \Gamma_q = \sum_k \frac{2g_k^2 \Gamma_{2,k}}{\Gamma_{2,k}^2 + \Delta_k^2} + \Gamma_q \qquad (4)$$

$$\Gamma_{\delta\delta} = \sum_k \frac{\Gamma_{qt,k}^2 \eta}{\Gamma_{t,k} + \Gamma_{qt,k}\eta}\left[\bar{Z}_H - \bar{Z}_L\right] \qquad (5)$$

where $\bar{Z}_H$ and $\bar{Z}_L$ are the time-averaged qubit polarization during the first half (reset to $|e\rangle$) and the second half (reset to $|g\rangle$) of the bath-polarizing cycles, $\eta$ is an interaction duty-cycle factor of the experiment protocol, defined as the percentage of time within a measurement block that the qubit-TLS interaction is active. $\eta < 1$ because the readout and reset pulses induce a significant Stark shift to the qubit, which renders its interaction with the relevant (near-resonant) TLS under study inactive. From Eq. (4), $\Gamma_\Sigma$ is a sum of the Lorenzians from discrete TLS and a small background decay rate $\Gamma_q$, which we assume to be a constant for simplicity. $\Gamma_{\delta\delta}$ can be closer to $2\Gamma_\Sigma$ with longer-lived TLS and better qubit initialization, and is closer to 0 if the TLS is short-lived. In Fig. 4(c), we show the results of this multi-TLS fit to $\Gamma_\Sigma$ and $\Gamma_{\delta\delta}$.

Many of these strongly-coupled TLS's lifetimes are in the millisecond range (Methods). We note that a strongly-coupled but short-lived TLS would register a strong peak in $\Gamma_\Sigma$ but show no response in $\Gamma_{\delta\delta}$. The linewidths of the TLS are on the order of a few MHz, reflecting their poor phase coherence and/or frequency stability, making the qubit-TLS interaction incoherent.

We have also briefly carried out two-timescale relaxometry on a fluxonium qubit in the planar circuit QED architecture. Here, we used QND measurement and real-time feedback for qubit reset instead of unconditional driven reset, as in the main device, but all other experimental steps are similar. This device shows about 12 prominent TLS over the frequency range of 125-325 MHz (see Methods). Most TLS display similar properties as in Fig. 4, i.e., a high degree of polarizability consistent with millisecond $T_1$ times and linewidths of several MHz.

### Dielectric loss analysis

Fluxonium qubits are more protected from dielectric loss than the industry-standard transmon qubits due to their significantly suppressed charge matrix elements, offset by a modestly higher surface participation ratio (SPR) from their smaller shunting capacitor. Following our standard practice of SPR simulation which treats the TLS bath as a continuum and excludes contributions from the junction oxide and surface layers near the small junctions[35], we find that the fluxonium qubit in this study has SPR about a factor of 4 higher than our typical 3D transmon designs (e.g. as used in Ref. [13]). This predicts $T_1$ times of about 2 ms assuming a typical surface dielectric quality for our fabrication process with one-step shadow-evaporated aluminum as in Ref. [35] (a 3 nm thickness surface layer with $\epsilon = 10$ and a weighted sum of loss tangent $\tan\delta = 2.6 \times 10^{-3}$), which roughly agrees with the background qubit $T_1$ time $1/\Gamma_q = 1.7 \pm 0.4$ ms in Fig. 4b, c.

However, the excellent background $T_1$ is barely relevant to the average-case performance of the fluxonium. Our observed discrete TLS resonances are not part of the dielectric surface layers in the aforementioned SPR modeling, but rather reside in the oxide tunnel barriers of the junction chain. Unlike a transmon, which consists of only one or two small junctions and statistically tends to encounter no resonant TLS in the junction(s) over a few GHz, the fluxonium consists of nearly 1000 times more volume of junction oxide and can expect no such luck. Compared to a TLS in the small phase-slip junction which would manifest itself as an avoided crossing in the qubit spectrum (with $g_k$ on the order of 10 MHz), a TLS in an array junction is estimated to have coupling $g_k$ that is several hundred times lower, resulting from a combination of the junction chain's voltage division effect and a suppressed charge matrix element (See Supplementary section VI). TLS in the surface layers of the capacitor pads and leads of the flux-onium would couple to the qubit too weakly to be individually observable. The observed density ($\rho = 40{-}60\,\text{GHz}^{-1}$) and qubit-coupling strength (20-80 kHz) of the TLS correspond to an area density of 0.3-0.45 $\text{GHz}^{-1}\mu\text{m}^{-2}$ and effective electric dipole moment of 2-9 Debye, assuming they are indeed dielectric TLS in the junction chain. Interestingly, these properties are very similar to those reported for TLS in aluminum oxide in the 3–10 GHz range that have been intensively studied in the past[7,36], suggesting a common origin across nearly two decades of frequency. This agreement, however, should be seen as plausible evidence rather than a direct proof that our observed TLS are of dielectric origin. In addition, our observation does not rule out the existence of other types of TLS with different properties in the same frequency range that may be invisible (i.e., much weaker coupled) to the qubit. On the contrary, the broad (~MHz) linewidths of the observed TLS may even be a hint to a denser bath of such invisible TLS, since TLS dephasing is believed to be caused by interacting with other thermally-activated off-resonant TLS.

## Discussion

Our results suggest that the fluxonium's coherence property is dominated by a forest of discrete and long-lived TLS residing in the oxide layers of the junction chain, with their average spacing only slightly larger than their linewidths. In retrospect, signatures of non-Markovian relaxation from TLS date back to the pioneering study of 3D fluxonium more than a decade ago[23], including multi-exponential qubit relaxation and up to 8 ms qubit $T_1$ measured with saturation pulses (which remains unmatched by the fluxoniums today and we now believe may be attributed to long TLS relaxation times). Previous studies of the fluxonium $T_1$ spectrum[37,38], including Ref. [17] on this very device, provided hints on this dense but individually resolvable TLS population from their large $T_1$ fluctuations versus qubit frequency. Our use of two-timescale relaxometry finally allows us to disentangle the long lifetimes of these TLS from qubit characterization, resulting in a clean snapshot of the TLS environment.

Looking forward, the junction chain quality will likely present a major roadblock to improving qubit coherence through circuit Hamiltonian engineering. The measured low background decay rates $\Gamma_q$ and the two-orders-of-magnitude variations of qubit relaxation rate from TLS resonances allows one to reconcile the record-setting fluxonium coherence times in best-case scenarios and the low quality factor of lossy junction oxide averaged over any practical spectral range (consistent with previous studies of Al/AlO$_x$ junction chain resonators[39,40]). The Josephson junction chain functions as a high-impedance superinductance, which plays a key role not only in fluxoniums but also in various protected qubits such as the 0-$\pi$ qubits[41], bifluxon qubits[42], fluxonium molecule[43], and a growing family of $\cos(2\varphi)$ qubits[44,45]. The relatively low average quality of the junction chain and the non-Markovian dynamics caused by a complex TLS landscape may have to be considered in these novel qubits analysis. Reducing TLS density in the AlO$_x$ junction chain or finding alternative low-loss superinductors[46] appears essential for systematic advances in qubit coherence.

On the other hand, our study suggests a few interesting directions to investigate and potentially exploit TLS properties. Our qubit-TLS interaction regime is already in the Purcell regime coveted in the roadmap laid out in Ref. [9], but strong dephasing of the TLS makes it difficult for our qubit to benefit from the long $T_1$ times of the TLS. It will be interesting to explore potential dynamic decoupling[47] and spin-locking techniques[48] to suppress the dephasing of the TLS. Furthermore, the fact that our observed TLS density remains comparable to previous literature values despite the much lower frequency and qubit-TLS interaction scale suggests a favorable trade-off to systematically explore even lower frequencies. Furthermore, we argue that our

**Table 1 | Device parameters**

| | Cavity parameters | | | Fluxonium parameters | | | | Freq. and coherence at half-flux | | | | | Readout and reset parameters | | |
|---|---|---|---|---|---|---|---|---|---|---|---|---|---|---|---|
| | $\omega_c/2\pi$ (MHz) | $\chi/2\pi$ (MHz) | $\kappa/2\pi$ (MHz) | $E_J/h$ (GHz) | $E_C/h$ (GHz) | $E_L/h$ (GHz) | Effective qubit temp. (mK) | $\omega_{01}/2\pi$ (MHz) | $\omega_{12}/2\pi$ (MHz) | $T_1$ ($\mu$s) | $T_2^*$ ($\mu$s) | $T_{2E}$ ($\mu$s) | Readout length ($\mu$s) | Readout photon | Reset length ($\mu$s) |
| 3D device | 7519 | 0.41 | 8.7 | 4.88 | 1.09 | 0.56 | 27 | 198 | 4430 | / | 320 | 670 | 15.4 | ~100 | 35 |
| Planar device | 6993 | / | 0.2 | 4.08 | 0.92 | 0.35 | > 100 | 130 | 3825 | / | 78 | 83 | 4.6 | / | 4.6 |

$T_2^*$ and $T_{2E}$ are measured when there are no apparent TLS strongly affecting the qubit at half-flux. $T_1$ is not listed as it is better represented by the two-timescale relaxometry data. The planar device has an equilibrium qubit population in $|g\rangle$ and $|e\rangle$ very close to 50%:50% at half flux.

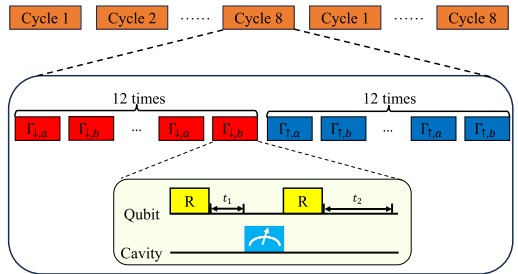

**Fig. 5 | Pulse sequence for two-timescale relaxometry with cycling delays.** This protocol variant is suitable for measuring qubit relaxation in the presence of a dominant long-lived TLS, using a cyclic arrangement of 8 relatively short delay times (CD-8), in contrast to a fixed arrangement of 4 delay times (FD-4, Fig. 3). Each sequence block contains only a single qubit readout following a variable delay time ($t_1$). To ensure that the TLS is consistently polarized as a function of block index $N$ for different cycles, the total qubit-TLS interaction time in each block ($t_1 + t_2$) is held constant. Furthermore, an alternating block structure (a, b, a, b,...) is introduced to equalize the variance of the qubit-TLS interaction periods in different cycles.

observation of $\Gamma_{\Sigma L} \approx \Gamma_{\Sigma H}$ (also see Ref. 12) suggests that $\Gamma_\Sigma$ is largely independent of the TLS bath temperature, and hence the Boltzmann temperature factor commonly accepted for qubit decoherence analysis[19,37,49] may not apply to dielectric loss from TLS, potentially making the heavy-fluxonium regime ($\omega_{01} \ll k_B T$) more favorable than previously appreciated.

Finally, the two-timescale relaxometry demonstrated here provides a general framework for an efficient probe of non-Markovian relaxation dynamics of any type of qubits. As long as the qubit can be initialized (without waiting for the bath to reach equilibrium), this technique can readily replace traditional $T_1$ measurements in qubit characterization routines, yielding similar measurement throughput while screening and characterizing potential non-Markovian behavior.

## Methods
### Device parameters
The fluxonium device in a 3D cavity is the same device as presented in Ref. 17. The qubit and cavity parameters characterized from our measurements are listed in Table 1, while information on device fabrication and packaging can be found in Ref. 17.

The planar device is fabricated and packaged by the Superconducting Qubits in Lincoln Laboratory (SQUILL) Foundry at MIT Lincoln Laboratory. It is fabricated on a 350 $\mu$m silicon wafer with a 200 nm base-metal layer of aluminum for large capacitor structures and 30 nm/160 nm double-angle evaporated aluminum for the Josephson junctions. The parameters of the qubit and the resonator are also listed in Table 1.

### Two-timescale relaxometry with cycling delay times
The two-timescale relaxometry protocol presented in Fig. 3 (a) consists of sequence blocks with 4 qubit readouts following four fixed delay times (0 $\mu$s, 40 $\mu$s, 150 $\mu$s, and 330 $\mu$s). These 4 readout outcomes are to be associated with the same bath variable $T$ under the approximation that assumes the bath is slowly varying. We will refer to this protocol as FD-4. For regimes where the bath contains a prominent single

TLS strongly influenced by the qubit dynamics, this approximation breaks down, and hence we use sequence blocks with only 1 qubit readout instead. The readout is carried out following a delay time $t$ that varies from block to block, cycling through a list of 8 different values (1 $\mu$s, 10 $\mu$s, 20 $\mu$s, 30 $\mu$s, 40 $\mu$s, 50 $\mu$s, 60 $\mu$s and 70 $\mu$s), which we will refer to as CD-8. Detailed pulse sequence is shown in Fig. 5. In our implementation, we loop through Cycle 1 to 8, and in each cycle we polarize the TLS to the high and low states by $N = 12$ sequence blocks each. The total qubit-TLS interaction time within each block is kept constant, so the bath state at the same $T$ (or block index $N$) for different cycles is comparable. Analysis of the relaxation rates is carried out in two steps, similar to the FD-4 experiment, but the first step is an 8-point exponential fit using all measurement outcomes for the same sequence block index ($N$) from all 8 cycles. More detailed modeling is described in Supplementary Section III(c).

As CD-8 is modestly more time-consuming than FD-4 and has less dynamic range to probe slow relaxation processes, we use FD-4 in general spectroscopy sweeps. When the extracted $\Gamma_\Sigma$ is larger than a threshold number (4 ms$^{-1}$ for the spectroscopy shown in the main text), we add an additional CD-8 experiment, and the result overwrites the FD-4 experiment's in spectroscopy for this flux point. This is carried out for more accurate estimates of individual TLS properties. For a heuristic view of the qubit $T_1$ spectrum with non-Markovian effects, the FD-4 protocol (Fig. 3a) would suffice.

### Relaxation spectrum of the 3D fluxonium device
Here we provide additional information associated with the relaxation spectrum presented in Fig. 4c. In Table 2 we present detailed results of all the TLS parameters, analyzed based on Eqs. (4), (5). Figure 6 shows the decay rates of a round-trip sweep that Fig. 4c is based on. The frequency sweep started from half-flux (198 MHz), went up to 393 MHz, and came back down, taking around 36 hours to complete (including about half the time spent on the refined CD-8 protocol for individual TLS analysis). The two sweeps have good consistency above 270 MH,z but some discrepancy below that due to TLS reconfiguration.

**Table 2 | Extracted parameters of individual TLS in the 3D fluxonium device from Fig. 4**

| TLS number | 1 | 2 | 3 | 4 | 5 | 6 | 7 | 8 | 9 | 10 |
|---|---|---|---|---|---|---|---|---|---|---|
| $g/(2\pi)$ (kHz) | 50 ± 2 | 57 ± 3 | 59 ± 3 | 71 ± 5 | 75 ± 3 | 32 ± 5 | 16 ± 7 | 30 ± 4 | 66 ± 2 | 27 ± 2 |
| $\Gamma_2/(2\pi)$ (MHz) | 1.6 ± 0.2 | 1.6 ± 0.3 | 1.8 ± 0.2 | 2.0 ± 0.4 | 2.0 ± 0.3 | 5.1 ± 2.2 | 0.2 ± 0.2 | 3.5 ± 1.3 | 2.0 ± 0.2 | 2.3 ± 0.5 |
| Frequency (MHz) | 204.9 ± 0.2 | 220.6 ± 0.2 | 240.6 ± 0.3 | 265.5 ± 0.3 | 278.0 ± 0.2 | 298.2 ± 1.2 | 305.3 ± 0.2 | 326.3 ± 1.0 | 349.2 ± 0.1 | 360.8 ± 0.3 |
| $\Gamma_t$ (ms$^{-1}$) | 0.6– 1.6 | 0.4– 1.4 | 0.2– 1.3 | <0.1 | 1.4–3.6 | ~10 | <0.1 | <0.1 | 1.5–2.6 | <0.1 |
| TLS polarizability (normalized) | 0.57 | 0.64 | 0.60 | 0.59 | 0.70 | 0.03 | 0.71 | 0.73 | 0.55 | 0.66 |

The analysis model is described in Supplementary Section III. Normalized TLS polarizability is calculated as $p_k^\Delta/(Z_e - Z_g)$ when it is on-resonance with the qubit, where $p_k^\Delta$ is the TLS polarization difference, approximately $\bar{Z}_H - \bar{Z}_L$ (for a more accurate expression, see Eq. (S13) in the Supplementary Materials), and $Z_e - Z_g$ can be interpreted as the maximum qubit polarization difference (determined by the reset fidelity). Their ratio is an indicator of the TLS lifetime. We note that the fitting result for TLS #7 has very large uncertainties as the peak is mostly composed of a single point, but this data point is prominent in both in back and forth sweeps, and hence is likely a discrete TLS with a narrow linewidth.

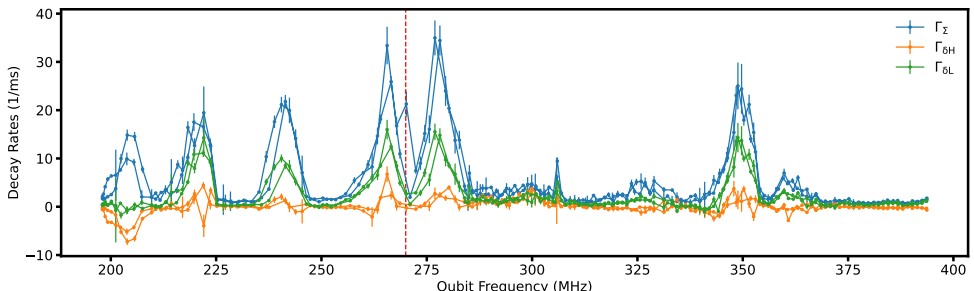

**Fig. 6 | Relaxation spectroscopy data of the 3D fluxonium device.** Qubit relaxation rates over a 36-hour round-trip sweep starting from the low end of the frequency range. The red dashed line separates the qubit frequency into two regions. The two data sets show good agreement in TLS resonance features for the higher-frequency region and some disagreement for the lower-frequency region due to TLS reconfiguration. The averaged data from both sets for the high-frequency region and only the return-sweep data set from the low-frequency region are included in the plot presented in Fig. 4c. Error bars represent one-standard-deviation uncertainty of fitted decay rates from the model described in the text.

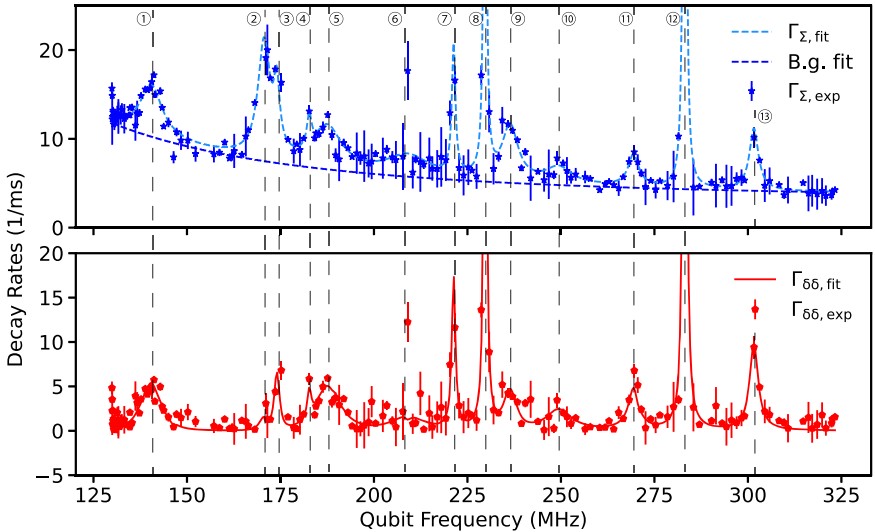

**Fig. 7 | Relaxation spectroscopy data of the planar fluxonium device.** This data set is acquired in a 15-hour frequency sweep with sub-optimal sequence parameters, hence it contains larger uncertainties than the 3D device. The measured $\Gamma_\Sigma$ and $\Gamma_{\delta\delta}$ are fit to our spectroscopy model, Eqs. (4) and (5) with 13 discrete TLS, together with a background qubit relaxation rate ($\Gamma_q$, dashed blue line on the top panel) that includes a frequency-dependent contribution from non-equilibrium quasiparticles, Eq. (6). Error bars represent one-standard-deviation uncertainty of fitted decay rates from the model described in the text.

## Relaxation spectroscopy of the planar fluxonium device

The long TLS lifetimes observed in the main device in this study and in Refs. 12,13 are somewhat surprising given that these devices do not have any engineered phononic structures to shield them from phonon-mediated decays (in contrast to Ref. 9,11). Coincidentally, these experiments are done in a 3D circuit QED device architecture, where the qubit chip is fully enclosed by a 3D cavity and only makes edge contact to the cavity package. To investigate the possibility that the long TLS lifetime may be related to any accidental phononic environment of a 3D cavity packaging, we carried out the non-Markovian relaxation spectroscopy of a fluxonium in a planar device.

This device contains a junction chain with 300 large Josephson junctions, each 200 nm in width and 3 μm in length. We employ our FD-4 protocol with four delay times 0 μs, 15 μs, 30 μs, and 50 μs. Qubit

**Table 3 | Extracted parameters of individual TLS in the planar fluxonium device from Fig. 7**

| TLS number | 1 | 2 | 3 | 4 | 5 | 6 | 7 | 8 | 9 | 10 | 11 | 12 | 13 |
|---|---|---|---|---|---|---|---|---|---|---|---|---|---|
| $g/(2\pi)$ (kHz) | 40 ± 3 | 45 ± 2 | 23 ± 3 | 15 ± 4 | 40 ± 5 | 34 ± 9 | 26 ± 5 | / | 35 ± 3 | 24 ± 6 | 22 ± 2 | / | 22 ± 1 |
| $\Gamma_2/(2\pi)$ (MHz) | 3.5 ± 0.4 | 2.0 ± 0.3 | 0.9 ± 0.3 | 0.8 ± 0.3 | 4.3 ± 0.9 | 7 ± 5 | 0.6 ± 0.4 | / | 2.5 ± 0.5 | 5 ± 2 | 1.9 ± 0.5 | / | 1.5 ± 0.4 |
| Frequency (MHz) | 140.4 | 170.6 | 174.0 | 182.7 | 187.6 | 209.05 | 221.3 | 229.7 | 236.4 | 249.4 | 269.4 | 283.1 | 301.6 |
| $\Gamma_t$ (ms$^{-1}$) | 1.0 ± 0.2 | ~75 | ~1 | ~1.5 | 0.5 ± 0.2 | ~2 | <0.1 | <0.1 | 3.5 ± 1.2 | <0.1 | <0.1 | <0.1 | <0.1 |

To contain the propagation of large uncertainties from this brief spectroscopy sweep, we manually fix all TLS frequencies $\omega_{t,k}$ by observation, and fit the other three parameters of each TLS. Note that TLS #8 and #12 have large uncertainties due to the lack of on-resonance data points, but their inclusion is helpful for fitting other TLS.

initialization is realized by measurement-feedback-based active reset with a reset fidelity of over 90% at half-flux. We observe approximately 12 discrete observable TLS in this spectroscopy (Fig. 7), corresponding to a TLS density of $0.33/\mu m^2/$GHz. Note that due to the larger number of junctions in the junction chain and the higher background decay qubit rate $\Gamma_q$, some features of discrete TLS in the junction chain may be more difficult to resolve from the background, so the actual TLS density may be larger. We attribute the high background qubit decay rate to non-equilibrium quasiparticles in the junction chain, partially motivated by the observation of high steady-state excited-state population of the qubit and an increased $\Gamma_q$ near half-flux. Therefore, the Markovian background of qubit $\Gamma_q$ is modeled as:

$$\Gamma_q(\omega_{01}) = 4|\langle 0|\hat{\phi}|1\rangle|^2 \frac{E_L}{h} x_{qp} \sqrt{\frac{2\Delta}{\hbar\omega_{01}}} + \Gamma_{q,0}, \qquad (6)$$

where $\Delta$ is the superconducting gap for Al, and we take it as $3.4 \times 10^{-4}$ eV here. Analysis applied to this spectroscopy indicates that most TLS intrinsic lifetimes vary from a few hundred microseconds to a few milliseconds (Table 3). The fit yields $x_{qp} = (2.3 \pm 0.1) \times 10^{-8}$ and a uniform background decay rate $\Gamma_{q,0} = (3.1 \pm 0.2)/$ms. Therefore, these results show that it is commonplace for the qubit-coupled TLS in the 100's MHz range to have long lifetimes in the phononic environment of planar qubits as well.

**Two-timescale relaxometry with variable polarity-switch structure**

In the measurement and analysis protocols of FD-4 and CD-8, non-Markovian effects from weaker-coupled TLS can be overshadowed by the presence of more dominant TLS. To reveal the impact of these less-coupled TLS on the qubit, in another variant of the relaxometry protocol, we skip the grouping process (as we do for FD-4 and CD-8) and track how the measured qubit population following a single fixed delay time $t$ changes as a function of the pattern of the initialization sequence. As we switch qubit polarization after a variable number of sequences, we call it "two-timescale relaxometry with variable polarity-switch structure", which enables us to go beyond the simple exponential model and directly probe more than one bath relaxation timescale $\tau_e$. Detailed description, modeling, and experimental results are presented in Supplementary section V.

## Data availability
The data used to reproduce the plots within the paper have been deposited in the figshare database under accession code doi.org/10.6084/m9.figshare.30727316.

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

## Acknowledgements

We thank E. Dogan for experimental assistance and Y.-Y. Wang for helpful discussions. This research is supported by the US Army Research Office, QC-S$^5$ Program (No. W911-NF-23-10093; Z.-T. Z., D. R., and C. W.) and the US Army Research Office, HiPS Program (No. W911-NF-18-10146; A. S. and V. E. M.). Data analysis is partially supported by the US Department of Energy, Office of Science, National Quantum Information Science Research Centers, Co-design Center for Quantum Advantage under contract DE-SC0012704 (Z.-T. Z., B.-J. L., and C. W.). The planar fluxonium device is designed by T. A. Masum, and it is fabricated and provided by the SQUILL Foundry at MIT Lincoln Laboratory, with funding from the Laboratory for Physical Sciences (LPS) Qubit Collaboratory.

## Author contributions

Z.-T. Z. carried out the measurements, collected the data, and performed the analysis with the assistance of D. R. and B.-J. L. The experimental protocol was initially developed by D. R. C. W., who conceived and supervised the experiment. The 3D device was fabricated by A. S. under the supervision of V. E. M. Z.-T. Z., B.-J. L. and C. W. wrote the manuscript with input from all authors.

## Competing interests

The authors declare no competing interests.
