## [Transparent Peer Review file · Nature Communications]

Non-Markovian Relaxation Spectroscopy of Fluxonium Qubits

Corresponding Author: Professor Chen Wang

Version 0:

Reviewer comments:

Reviewer #1

(Remarks to the Author)
see reviewer attachments.

Reviewer #2

(Remarks to the Author)

The authors propose and experimentally demonstrate a technique for studying two-level systems that are weakly coupled to a qubit, as well as their effect on qubit relaxation. A key feature of the method is that the environment initialization is performed simultaneously with the qubit relaxation measurement, which significantly reduces the total measurement time compared to conventional approaches.

The manuscript is well written, and I believe the proposed relaxometry approach will be useful for studying noise sources, improving device characterization, and developing more coherent qubits. However, I recommend the authors address the following points.

Major points:

The authors mention that, in conventional relaxation measurements, the results can depend on the choice of time step, reset protocol, and averaging techniques due to non-Markovian effects. I believe a similar analysis should also be provided for the data obtained using the proposed technique. In particular, it would be helpful to clarify how the choice of time step and time range affects the extracted quantities, especially considering that the derivative is determined from an exponential fit to $Z(t)$, while the actual Z dependence on time differs from a purely exponential form.

The authors should be more careful in citing prior work (lines 15–16). The phrase “studied in relatively contrived demonstrations” seems inappropriate, as non-Markovian noise has been investigated in real devices. For example, studies on quasiparticles [S. Gustavsson et al., *Science* 354, 1573–1577 (2016)] and near-resonant TLS and TLFs [J. Burnett et al., *npj Quantum Information* 5, 54 (2019)] have explored non-Markovian effects in realistic experimental settings. The statement in lines 219–222 is also inaccurate in light of [J. Lisenfeld et al., *Nat. Commun.* 6, 6182 (2015)] and other related works. I recommend the authors revise these formulations and include appropriate references.

Minor points:

It may be helpful to indicate the corresponding peaks from Extended Data Table 2 directly in Figure 4(c), for example with numbers or vertical dashed lines, as some are difficult to distinguish. This would improve readability.

In lines 198–199, the authors argue that the observed similarity between the relaxation rates in the low and high environment states ($\Gamma_{\Sigma L} \approx \Gamma_{\Sigma H}$) indicates the absence of non-Markovian effects from a bosonic bath. However, this conclusion is not entirely evident. Could the authors clarify how this similarity excludes non-Markovian contributions from bosonic environments? A more detailed justification would strengthen the argument.

The reset fidelity in the 3D architecture shows significant variation. Could the authors explain why this reset method was chosen over the one used in the planar device, particularly since the proposed relaxometry relies on efficient initialization?

The T_2 and T_2E values were measured when TLS did not strongly affect the qubit. Could the authors clarify how this condition was ensured? Was it achieved by detuning the TLS from the qubit or by excluding the data points where strong coupling was observed?

Supplementary, line 108: For improved readability, I suggest replacing the reference to Eq. (S23) with a reference to Eq. (S5).

Supplementary, lines 177–178: The authors attribute the observed readout phase difference at low pumping to residual reset effects from the previous pulse. Why does this effect appear only at frequencies above 380 MHz?

Figure S8 caption: There appears to be an inconsistency – ‘1’ and ‘0’ should likely be replaced by ‘↑’ and ‘↓’. Also, the phrase “pumping and initialization are in the same direction” may be misleading. A clearer version could be: “The ‘↑’ initialization requires only pumping, while the ‘↓’ initialization requires pumping followed by an additional π pulse.”

I believe the manuscript presents a valuable experimental approach with potential impact on noise characterization. Provided the authors address the points raised above, I would support the publication of this manuscript.

Reviewer #3

(Remarks to the Author)

The authors explore the relaxation of a fluxonium qubit and a 3D cavity and find that it does not fit the usual Markovian assumptions with simple T_1/T_2 times. This is shown to be due to coupling to a TLS bath, and the dynamics of interaction and joint relaxation of the fluxonium qubit and the TLSs are explored. The authors introduce a new protocol to efficiently study this process, where the qubit relaxation is probed during a long polarization of the TLS bath to the excited and ground states. The authors show quantitative agreement with a simple model for a qubit coupled to several TLS defects, and are able to extract relevant rates for the decay and interaction of the systems. Remarkably, by varying the fluxonium frequency in the low frequency regime 0.1-0.4 GHz, the authors observe a significant change in the relaxation dynamics - and show they can be related to several resonant TLS defects coming to resonance with the qubit in that frequency regime. The manuscript is clearly written and well explained. We believe the manuscript clarifies a mystery behind the lifetimes of fluxonium qubits and presents a new way to quantify the TLS bath coupled to it. Thus, it is relevant to the general community studying the coherence of superconducting circuits, and the more fundamental study of interaction with a spin bath in general. We believe it is suitable for publication in Nature Communications.

We have several questions that arose while reading the manuscript, which can hopefully be better explained:

1. In fig. 3d, the authors mention that the fast relaxation is not due to the intrinsic TLS lifetime being shorter, but rather due to stronger interaction with the qubit. This can also be seen in the initial rates in Fig. 3d being significantly higher than those of Fig. 3c. In that case, is the TLS relaxation rate just given by the background decay rate of the fluxonium qubit (without TLS) or is it a more complex interaction?

2. Can these measurements also be used to extract the intrinsic TLS lifetime (due to e.g. phonons) reliably? This can be particularly interesting in systems such as Ref. 22 where the phonon bath can be tuned.

3. The observation that the decay rates at positive and negative polarization are equal ($\Gamma_{\sigma_H} \sim \Gamma_{\sigma_L}$) implies that the temperature of the TLS bath does not affect the dynamics. However, the authors also comment that it indicates "an absence of non-Markovian effect from any bosonic bath". This statement should be clarified. Which bosonic bath is assumed and why is this seen as an absence of it. In general, is it clear that these defects which interact with the qubit have only two levels and not more?

4. This manuscript focuses on defects in the AlOx layer as the main source of TLS, while previous work such as Ref. 20 discussed them more generally without being certain of their origin. Specifically, magnetic defects were also considered as options. Would this also be a possibility in this experiment or can it be excluded? How would the authors explain the change of the TLS spectrum with time (Fig. 4b), which perhaps seems more reasonably due to magnetic defects?

5. In the model that included Γ_{σ} and Γ_{Δ} (Eq. 3,4), a duty-cycle factor was introduced because the interaction was not on for the entire duration. Why does this only affect the Γ_{Δ} term and not the general decay term Γ_{σ} ?

Reviewer #4

(Remarks to the Author)

I co-reviewed this manuscript with one of the reviewers who provided the listed reports. This is part of the Nature

Communications initiative to facilitate training in peer review and to provide appropriate recognition for Early Career Researchers who co-review manuscripts.

Version 1:

Reviewer comments:

Reviewer #1

(Remarks to the Author)

This revised manuscript is in my view a significant improvement over the previous version. The authors have addressed all the questions in my original review. I therefore recommend that the manuscript is published in Nature Communication.

However, I would like to insist on two minor changes:

- i) Regarding the new plots Fig. 2b, c, please set the y-axis on the same range. In this way one can see more clearly that the non-polarized T1 starts at $\sigma_z=0.4$, which coincidentally is the point of inflection in panel b. If the data can be compared to Fig. 4 (measured roughly at the same time), it would be very valuable to add the qubit frequencies to panel b, c.
- ii) The authors forgot to update the citation (s. comment 20).

FYI reply to comment 5b:

I have realized that my remark on the qubit reset distribution was premature and conflated with the issue of state-preparation errors in the active reset. However, in your reply to comment 5b, you implicitly assume that the qubit reset distribution is independent on the qubit population before the reset. Any such dependence can be interpreted as a state preparation error in your protocol. However, this error seems to be small in your experiment, as can, to some extent, be verified from Fig. S8.

Reviewer #2

(Remarks to the Author)

The authors have carefully addressed all of my comments and questions. The revised manuscript has been significantly improved, and the clarifications and additional explanations resolve my previous concerns. I find the responses to be clear and satisfactory. I recommend the manuscript for publication.

Reviewer #3

(Remarks to the Author)

The authors have made significant corrections and improvements in the revised version, and I believe they have addressed the questions and issues raised by myself as well as the additional reviewers. I do not have any further comments, and happily recommend this manuscript for publication in Nat. Comm.

Reviewer #4

(Remarks to the Author)

Summary of changes

- 1.** We added descriptions to specify the definition of “non-Markovian” in this paper both at the beginning of the introduction part and in section A.
- 2.** We acknowledged a few more prior experimental studies on non-Markovian noise and rephrased the corresponding introduction accordingly.
- 3.** We rephrased the sentence “This technique requires efficient reset of the qubit but does not require high-fidelity quantum non-demolition (QND) readout” to acknowledge protocols described in “Two-level system hyperpolarization using a quantum Szilard engine” while explicitly highlighting the generality of our protocols in the introduction.
- 4.** We rephrased some sentences both in the last paragraph of introduction and the end of section E to acknowledge that this study does not differentiate microscopic mechanisms of TLS (even though the result is well interpreted as junction-chain dielectric TLS).
- 5.** We listed more device parameters values at the beginning of section A in the main text, which was described in Table 1. We added more readout pulse information there as well.
- 6.** We revised Fig. 1(c) to make the plot look cleaner.
- 7.** We revised Fig. 2(c) by choosing a different example data set that can better demonstrate a non-polarizable case of the environment. Lines and data points formats have been updated for better visualization.
- 8.** In the caption of Fig. 2 we added a citation for directing readers to detailed derivations, and corrected an error of the first timescale.
- 9.** We added information on qubit reset for the 3D device close to the end of section A, which was originally described in “Methods”.
- 10.** Fig. 4(a) has been updated as we applied a more robust method to evaluate uncertainty of the qubit decay slope and included more data points.
- 11.** We revised Fig. 4(c) for correcting a mistake when taking the average for two datasets. Some extracted TLS parameters have been slightly changed, as shown in Table II.
- 12.** We moved the “Relaxation rate analysis” section from Method to the main text (section C).
- 13.** We added more explanations at the beginning of section D on why we think in our experiment bosonic environment has relatively small contribution to non-Markovian effect. Due to data quality limitations, we have slightly weakened our statement.
- 14.** When recapping previous qubit T_1 spectroscopy studies in section D, we acknowledged that coherent qubit-TLS interactions observed previously are also non-Markovian (but different from what we discuss here). We thus rephrase our sentence to make the statement more clear.
- 15.** We moved the description of the qubit reset method for the planar device from Methods to main text (at the end of section D).
- 16.** In Methods, other than moving some parts to the main text (as listed above), description of fridge diagrams and cryogenic setups is moved to the first section in the Supplementary Materials, and analysis related to surface participation ratio is moved to the last section in the Supplementary Materials.
- 17.** We added the readout and reset parameters in Table I, with more descriptions in captions instead of footnotes.
- 18.** We replaced the original “TLS polarizability spectrum” plot in a Extended Data Figure with TLS polarizability data in the last row of Table II.

19. We made a more explicit distinction between the situation where we used CD-8 versus FD-4 at the end of “Methods”, Two-timescale relaxometry with cycling delay times.
20. We revised the last sentence of the “Methods”, Relaxation spectroscopy of the planar fluxonium device, to soften the original wording and avoid making an overly strong claim about predicting long TLS lifetimes in the 100-MHz range.
21. We added the model description about how we extract the initial decay slope at the end of Supplementary section III(A).
22. We moved the claim that $\Gamma_{qt}^{eff} = 0$ from the end of Supplementary section III(C) to section III(B) and simplified formulas accordingly, so that the derivations are easier to follow.
23. Time-averaged qubit polarization expression is added at the end of Supplementary section III(B).
24. We added more explanation on how we choose delay time step t_1, t_2, t_3 and time range in Supplementary section III(D). Motivation about delay time step choice is also more explicitly mentioned in the main text section B.
25. We added a new subsection titled “*Verification of methods via full dynamics simulation*” (Section III(E)) to illustrate the performance and reliability of our protocol and parameter-extraction procedure when tested on synthetic datasets produced by full dynamical simulations.
26. We removed a supplementary figure which showed sample data of qubit reset calibration originally in Supplementary section IV “Calibration experiments”, because the information is already included in Fig. S5.
27. Fig. S5 has been updated to use qubit σ_z instead of phase contrast as z-axis to give readers more information. A new set of data is used here with a larger frequency range.
28. Fig. S6 and S7 have been updated for better visualization.
29. Fig. S9 showing “qubit initialization performance in practice” has been updated to show initialization for first and last polarization sequences as well as different protocols.
30. We changed a few notations to keep better uniformity in the paper. Most Italic subscripts have been corrected except for index.
31. Due to changes in article structure, some references to Method/Supplementary are changed/deleted.
32. We added a sentence to clearly describe how we perform decay rate analysis to extract qubit and TLS parameters at the end of Supplementary section III(C).
33. Other minor changes that were called by the reviewers and for better presentations.

Reviewer #1 (Remarks to the Author):

Comment 1: In this work, the authors prove the existence of long-lived two-level systems in state-of-the-art Josephson junction array (JJA) fluxoniums. These (most likely the same) long-lived TLSs were first discovered and characterized in granular aluminum. However, many questions remained open, especially regarding the nature of these TLSs, to which extent they are specific to the disordered nature of granular aluminum, and if these TLSs are also relevant for junction array fluxonium and other superconducting qubits.

While, in retrospect, there are indications of their presence in earlier JJA fluxonium experiments, as the authors also state, the results presented in this manuscript as well as the results of the Munich group presented at the March meeting, are the first and eagerly awaited scientific documentations of these TLSs in junction array fluxoniums.

The high TLS density in frequency and their frequency reconfigurations over time, that the authors extract, makes these TLSs a highly relevant loss-mechanism for the qubit, which means that strategies have to be found to mitigate the detrimental influence of these TLSs to further improve fluxonium qubits.

Strong arguments are provided (coupling strength and density in frequency) that these TLSs are of dielectric nature. Note, that in a recent experiment (arXiv:2501.03661) a magnetic origin could be ruled out.

Reply: We thank the Referee for affirming assessment of our work as one of the first and eagerly awaited scientific documentations of these TLSs in junction array fluxoniums.

Comment 2: Except for the conclusion, which is spot on and highlights the importance of the findings for the community, the rest of manuscript seems to be written in a rush. Important information is either absent, fragmented across different sections of the main text, methods and supplement, or must be inferred from an earlier work.

Unfortunately, I have to say that in the current form the manuscript does not meet the quality required for a high-profile scientific publication. Nevertheless, on the basis of the experimental results I highly recommend reconsidering a revised version of the manuscript for publication in Nature Communications.

Reply: We appreciate the Referee's thorough review of the manuscript. We have carried out various changes to improve the clarity and completeness of the manuscript. We believe the revised manuscript is suitable for publication in Nature Communications.

Comment 3a: Below I list 3 major issues I strongly recommend addressing followed by a list of detailed comments and recommendations

It almost seems as if deliberate effort went into hiding the incredibly slow readout and reset of the qubit (readout $15\mu\text{s}$ compared to typically $\sim (0.1-1)\mu\text{s}$, reset $35\mu\text{s}$). The fact that the reset is achieved with highly demolishing readout pulses is only mentioned in the methods section. The unusual readout pulse length is only mentioned in the supplement. No demolition errors are quantified. From the main text the reader can only speculate that the readout is not single-shot, which is highly confusing when the protocols show qubit readout and reset.

Reply: While we agree that having fast single-shot readout is a necessary qualification for building “good qubits” for quantum computing, the focus of our manuscript is not to showcase all-around operational metrics of our qubits, but rather on 1) the protocol to study qubit relaxation in the presence of environment memory effect, and 2) observation of discrete long-lived TLS in a new regime. The length of the readout and reset is neither a fundamental hindrance to our experimental implementation (so long as they’re much shorter than the millisecond bath relaxation timescale) nor an essential element for the general reader to understand our results/conclusions. Whether our readout and reset is “incredibly slow” or not is not particularly relevant in the scientific context of the paper.

Contrary to what the reviewer has implied as a negative, the fact that we carried out this study without single-shot QND readout is exactly what makes this protocol generally adaptable. In the revised introduction, we further emphasized the key attributes of the protocol in the introduction:

“It can tolerate large infidelities in qubit readout and reset assuming they can be calibrated...”

We also explicitly stated that the readout is destructive in Section A.

Finally, we should point out that QNDness of the readout is a completely irrelevant factor in our experiment, which is a key difference between our work and Spiecker et al., Nat. Phys. 2023. We will further discuss that in later point-by-point replies.

Comment 3b: Given that the readout plays a central role in the protocol and the analysis, I recommend adding the following information in the main text:

- The readout and reset length and photon number, the dispersive shift and linewidth of the resonator. Besides giving a better understanding of the system, it proves for example, whether the cross-relaxation is indeed inactive during the readout and reset, as the authors claim.

Reply: We thank the reviewer for the suggestion. We appreciate that from a superconducting qubit operation expert’s perspective that it is helpful to know the readout parameters upfront for easier comparison with other studies. Hence, we have now included these information in the main text section A:

“ $\kappa/2\pi = 8.7$ MHz, and $\chi/2\pi = 0.41$ MHz’,

‘Due to the large κ/χ ratio and the absence of a quantum-limited amplifier, qubit readout is carried out with a 15 us square pulse with approximately 100 average photons, which is often destructive (i.e. non-QND) to the qubit state’,

‘Here, the qubit reset is realized with a strategy of applying a 35 us intentionally non-QND readout pulse to the cavity [28], giving reset fidelity in the range of 60% to 95% (mean reset fidelity 72%’”.

We have also added some readout and reset parameters in the extended data table 1.

	Cavity parameters			Fluxonium parameters				Freq. and coherence at half-flux					Readout and reset parameters		
	$\omega_c/2\pi$ (MHz)	$\chi/2\pi$ (MHz)	$\kappa/2\pi$ (MHz)	E_J/h (GHz)	E_C/h (GHz)	E_L/h (GHz)	Effective qubit temp. (mK)	$\omega_{01}/2\pi$ (MHz)	$\omega_{12}/2\pi$ (MHz)	T_1 (μ s)	T_2^* (μ s)	T_{2E} (μ s)	Readout length (μ s)	Readout photon	Reset length (μ s)
3D device	7519	0.41	8.7	4.88	1.09	0.56	27	198	4430	/	320	670	15.4	~100	35
Planar device	6993	/	0.2	4.08	0.92	0.35	>100	130	3825	/	78	83	4.6	/	4.6

TABLE I. **Device parameters.** T_2^* and T_{2E} are measured when there are no apparent TLS strongly affecting the qubit at half-flux. T_1 is not listed as it is better represented by the two-timescale relaxometry data. **The planar device has an equilibrium qubit population in $|0\rangle$ and $|1\rangle$ very close to 50%:50% at half flux.**

On the other hand, we'd like to clarify that readout with better contrast and shorter length can make the experiment easier in the sense of reducing noise, but readout does not play a central role in the protocol and the analysis.

As shown in the table, the qubit frequency is shifted by ~40 MHz during readout, resulting in a residual Γ_{qt} of less than 0.1/ms with common TLS parameters we obtain. As the cavity's lifetime is less than 20 ns, time for cavity photons to reach equilibrium is also negligible. In short, treating qubit and TLS as decoupled during readout and reset is a proper assumption to the extent of our analysis precision.

Comment 3c: - The modeling of the protocol as well as the populations are currently based on the assumption that the readout is free from demolition effects. These effects are crucial for the achievable TLS populations and estimation of the TLS lifetimes. Thus, there should be a number in the main text that quantifies the demolishing errors of the readout (with details in the appendix). The authors may use a similar protocol as in Ref. 20.

Reply: In fact, the modeling is not at all based on the assumption of non-demolition readout. Demolition effect has two consequences: one is to contribute to measurement errors, which has been taken into account in the calibration of qubit readout (equivalent to the common use of a confusion matrix); the other is to leave the qubit in a different state than the measurement outcome, which plays no role in our experiment since the readout is immediately followed by an unconditional driven reset.

Comment 3d: - Similarly, I am expecting a plot in the appendix that shows that the reset is more or less independent of the TLS polarization. This plot can be compiled from the current data. That the readout is not free from memory effects is mentioned in the supplement. These state preparation errors of the autonomous reset can drastically influence sensitive parameters like Γ_t . I also want to note that since the reset is most likely not pure, i.e. the qubit population fluctuates from reset to reset and gives on average a ~70% fidelity, there will be deviations from the average dynamics, since the qubit and TLS populations are correlated. To make this point clearer: in the case of active resets, a faulty active reset has a strong influence on the dynamics that limits the polarizability more than proportionally.

Reply: As mentioned in Supplementary Section III C, in the CD-8 protocol (which is used to measure strongly-interacting situations), we included in our model to consider that the reset fidelity is dependent on TLS polarization. In FD-4 protocols (which is used when the interaction is not as strong), we treat the reset fidelity as independent of TLS polarization to improve statistical uncertainty. Upon the reviewer's request, for both protocols we have now included data of $|e\rangle$ and $|g\rangle$ state initialization from the first and last polarization sequence

as shown below (also in Supplementary section IV). This figure replaces the original figure that shows averaged qubit initialization spectroscopy in the supplementary.

FIG. S8. **Qubit initialization performance in practice.** Red and orange data points correspond to $|e\rangle$ state initialization, while blue and purple traces show $|g\rangle$ state initialization. For FD-4 protocols, the initialization roughly stays the same for first and last sequence block and in CD-8 there is a more prominent difference which can be explained by TLS affecting qubit state during idling time.

On the second question about state preparation error, it is similar to the last question of comment 5, so we have the explanation and derivation there. In short, according to our model, the average reset fidelity does affect the polarizability of the TLS, but an averaged treatment is valid.

Comment 4a: In my opinion the main finding of the manuscript consists in the observation of long lived two-level systems in standard aluminum qubits, not only in granular aluminum as first reported. I recommend, positioning the article in this context of the current literature. In contrast, the authors currently focus the article on the “introduction” of relaxation protocols, which are not necessarily faster and exist already in Ref. 20 together with a more profound analysis. The authors should work out more precisely the difference of their protocols and also compare their experimental findings with the results in Ref. 20.

Reply: We agree with the Reviewer that we had neglected a careful comparison of the measurement protocol with Specker et al Nat. Phys. 2023 (Ref. 20 of our previous version). Indeed, while the protocol of Fig. 3 and 4 in Ref. 20 follow the structure of “preparing the bath and then measuring qubit for a long time”, Fig. 2c in Ref. 20 indeed shares a similar principle as our protocol and is just as fast. In the revised text, we have now prominently acknowledged that experiment introduced in Ref. 20 in the introduction section:

“(Specker et al) also introduced a powerful technique dubbed the “quantum Szilard engine” that uses fast feedback control of the qubit to manipulate the bath while using the feedback-control records to characterize the dynamics of the joint system.”

To acknowledge the concern of the Reviewer, in addition, we have restructured the introduction to downplay the data throughput rate of our protocol. The old paragraph in the

introduction discussing that traditional probes of non-Markovian relaxation were slow has been eliminated altogether.

With that said, our technique is introduced in the context of revisiting the T_1 relaxation experimental protocol, which is a different context versus quantum-jump tracking and feedback control (as Ref.20). Although quantum-jump tracking in principle can also be used to extract T_1 values, there is a reason that people don't use it for long-range spectroscopy sweeps to study qubit relaxation (e.g. as a function of flux): It requires tuning up high-fidelity QND readout, and even a small degree of MIST effect (demolition effect from the readout pulse) poses a tradeoff vs measurement cadence, and the extrinsic MIST effect has to be carefully disentangled from intrinsic qubit relaxation rates. On the other hand, people have been measuring qubit T_1 decay curves long before single-shot QND readout was possible, and they will continue to do so. It is fair to say that our work introduced a measurement and analysis framework on how to do T_1 -relaxation type of measurements in the presence of long-lived environment memory effect with arbitrary (calibrated) readout and reset fidelity. This is particularly relevant in light of widely-existing misinterpretation of naive T_1 measurements .

Finally, we'd like to point out that the beautiful technical quality of Ref. 20 does not take away the novel aspects of our experiment. We have no doubt that the pristine readout fidelity and QNDness allows for a more "direct" and transparent interpretation of the decay rates, which holds a strong scientific appeal of its own. The value of a more "convoluted" protocol that can handle substantial imperfections under practical constraints is also worth appreciating, just in a different way. While the experiment in Ref. 20 shines in its readout and feedback performance, our experiment stands out technically in its record qubit (background) lifetime of 2ms, allowing for probing decay rates with nearly two orders of magnitude contrasts.

In short, while we regard Ref. 20 as truly great work, our protocol can be more widely adopted in devices that cannot guarantee clean, high-fidelity quantum non-demolition readout for a wide range of frequencies, which is not rare at all compared to the widely-exist qubit lifetime misinterpretation situations. It is true that our proposed protocol is not necessarily faster than the one discussed in Fig. 2(c) of Ref. 20 and we have made the comparison more explicitly in the introduction part of the revised manuscript. Detailed discussion is listed in following point-by-point replies.

Comment 4b: The discussion whether or not the dynamics is non-Markovian is quite subtle. In the field of quantum master equations, rate equations like the Solomon equations are considered as Markovian. In fact, the Solomon equations are based on the Markov approximation (the dynamics is CP-divisible, i.e. the qubit dynamics is governed by a time-local, but time-dependent master equation). Instead, the Born approximation is not fulfilled (Ref. 25). I recommend speaking of multiexponential relaxation and the memory: e.g. in the abstract: under Born- Markovian assumptions and could mask such non-exponential I would also suggest a new title in the following direction: Relaxation spectroscopy of long-lived TLS in Junction-Array Fluxoniums. (In my opinion, it is quite unfortunate that non-Markovian made it to the title of Ref. 22. If the authors wish to adhere to the non-Markovian nomenclature, I would recommend clarifying what is meant by it.)

Reply: We thank the Referee for the detailed comments on the subtlety of the definition of non-Markovianity. It is true (and unfortunate) that there exist different definitions of Markovianity that are not fully compatible. While the Solomon-equation dynamics does not violate a quantum information theoretical definition of Markovianity (CP-divisible, no information back flow), it does violate the definition of Markovian (not only violate Born) in the true classical definition of Markovianity—future evolution of the state only depends on its current state—which we would argue should prevail in terms of a longer history, wider adoption beyond QIS, and easier to interpret. Therefore, we will maintain the current title of the manuscript.

Nonetheless, we agree that the reviewer’s concern is valid, and we have made the changes in the introduction and Section A (right after Solomon equations) to define more explicitly what we mean by non-Markovian:

“...qubit relaxation becomes apparently non-Markovian as it depends on its past operation history.” (introduction)

“A general solution to these (Solomon) equations leads to non-Markovian qubit relaxation dynamics in the sense that it violates the classical definition of Markovian processes [28], although certain quantum information theoretical definitions of Markovianity may differ (e.g. [29]).” (Section A)

We also made some edits in the abstract in line with the reviewers:

“However, the standard techniques used to characterize qubit relaxation is only valid for measuring T_1 under the Born-Markov approximation ~~assumptions~~ and could mask ~~non-Markovian behavior from the environment~~ environmental memory effect in practice.”

Comment 4c: The different protocols that are “introduced” are essentially a consequence of the challenging readout. The slowly varying qubit transition times as a function of time T , on which the paper is based on, can be seen in Fig. 2c of Ref. 20. Since I do not know of any other device in the community that is read-out that slowly, I highly doubt these specific protocols are relevant for the community. With a standard readout (single-shot and >90% fidelity) one would rather want to reset the qubit actively with a constant repetition time (as is done in Ref. 20). In this way, one can correct for SPAM errors and demolishing effects (Ref. 20).

Reply: It is true that our protocol shares conceptual similarity with the approach demonstrated in Fig. 2(c) of Ref. 20. In the revised manuscript, we now make this comparison explicit at the end of the second paragraph of the Introduction. Although our demonstrations were performed on a device with challenging readout, this in fact illustrates the robustness of our protocol (as discussed earlier in Comment 3(a)) rather than limiting its applicability. The protocol is not restricted to slow-readout devices, nor does slow readout diminish its relevance. We thus respectfully disagree with the assertion that the community would not adopt our protocol due to readout speed. Readout speed is not the determining factor in the scientific context of this work.

On errors analysis, quantum-jump tracking methods are intrinsically susceptible to QND infidelity (since a readout-induced flip can be confused with an intrinsic flip for the following readouts) whereas our T_1 -like slope-based measurements (i.e. reset  variable delay  measure) and analysis do not suffer from this sensitivity. In our analysis, SPAM

errors are corrected using the calibrated qubit initialization average populations P_g and P_e . As discussed in Comment 3(c), the demolition effects do not influence our analysis since each measurement is immediately followed by an unconditional reset.

Comment 4d: The only addition in the protocol presented here compared to Ref. 20 is that cooling and heating sequences are alternated instead of looking at the qubit relaxation in between, as in Ref 20. The total measurement time is not necessarily faster. In contrast to the protocol in Ref. 20, following the protocol proposed in this manuscript, one loses the cleaner relaxation data, which is free from state preparation errors, as well as the important information of the “heating + qubit relaxation from ground state” and “cooling + qubit relaxation from excited state” data. In both protocols, the data throughput is the same and essentially depends on the measurement repetition time. Instead, I think the only true advantage in this manuscript comes from alternating the polarization, which confines the dynamics more towards the most resonant TLSs. The paragraph ll. 126-133 is thus a bit misleading: 1. data is acquired during the polarization sequence to compute the transition rates. 2. the length of the polarization sequence must in any case be adapted to the millisecond time-scales Γ_{qt} of interest. In the present manuscript the authors adapt the protocol on the fly and remeasure. In short, I recommend to differentiate the protocols more precisely, taking into account the arguments above and directing the focus more towards the experimental findings.

Reply: We acknowledge that our description of the protocol as “more efficient” was meant specifically in comparison to the widely used “bath-prepare \rightarrow delay \rightarrow probe” schemes (as in Fig. 2 of our manuscript, Fig. 3 of Ref. 20, and many related studies). We had not adequately contrasted our protocol with Fig. 2(c) of Ref. 20 in the Introduction, and we have now explicitly mentioned this method protocol of Ref. 20 in the introduction, and clarified the comparison in terms of the advantages of our approach—namely, that “it can tolerate large readout and reset infidelities provided that they can be calibrated”.

While the relaxation data in Ref. 20 is visually cleaner than ours, it is not completely free from state-preparation errors. Any non-QND component must be carefully separated from genuine jumps due to relaxation in their method. In contrast, although our data appears ‘dirtier’, they remain fully analyzable and do not place stringent demands on QND performance. Our parameter extraction does not require information from the separate ‘heating + relaxation from ground state’ and ‘cooling + relaxation from excited state’ curves.

Regarding the description in lines 126–133, this text pertained specifically to the experiment in Fig. 2. In that experiment, data are indeed taken only after the initial polarization sequence and the subsequent delay, not during the polarization itself.

Comment 4e: Throughout the paper the long-lived TLSs are implicitly assumed to be “conventional” dielectric TLSs. ll. 694ff. “Therefore, these results show that low frequency TLS generally have long lifetimes in the phononic environment of planar qubits as well”. In my opinion, statements like this are too much of an oversimplification, given the fact that the microscopic origin of dielectric TLSs is still mysterious with likely multiple coexisting TLS classes (review Ref. 21 cited by the authors). The fact that the life-time is several orders of magnitude higher than what is usually observed is a strong conflict that can hardly only be explained by the frequency. Note that these long-lived TLSs are still observed at a few GHz

in granular aluminum. Just to put these lifetimes into perspective, the phononic-bandgap engineering of Ref. 22 could tremendously enhance the TLS lifetimes from hundreds of ns to only tens of μs , with three exceptions on a $\sim\text{GHz}$ span with qubit relaxation in the ms-range. Another exotic TLS with a long lifetime was reported by two of the authors with a transition frequency of 2.9 GHz. These findings suggest that we should not lump together all these TLSs. The authors may for example use the opportunity to revive the ideas of Schechter & Stamp suggesting two TLS classes (mentioned also in Ref. 21), one at lower frequencies with long lifetimes due to a vanishing coupling to phonons, while the other at higher frequencies has much shorter lifetimes. Regarding the proposed comparison to previous findings, using the measured TLS coherence time of $\sim 1.5\text{MHz}$ yields a similar coupling strength $g = 2\pi \cdot 24\text{kHz}$ for the TLSs in granular aluminum but a higher density of ~ 1 TLS per MHz, which might be expected for a disordered material.

Reply: In this paper we mentioned that ‘we find that the relaxation spectrum is well explained by a random distribution of prominent TLS in the aluminum oxide tunnel barrier of the fluxonium’s Josephson junction chain together with a much weaker continuum of surface dielectric loss (on the order of ms^{-1}).’ It is not providing an absolute rebuttal to multiple-type of TLS, but only suggesting all current findings do not provide contradiction to what we think of conventional dielectric TLS, as observed TLS density, qubit-TLS coupling and TLS lifetimes can be somewhat expected with dielectric based analysis. While we know that some long-lived TLSs are observed in GHz range as the reviewer also mentioned, a typical estimation of TLS lifetime for 3-10 GHz frequency range would still be on the order of 100ns. If we assume TLS lifetime is limited by coupling to 3D acoustic phonons, for low temperature cases ($\hbar\omega \gg k_B T$) Γ_t is roughly in proportional to ω^3 according to Fermi’s golden rule (see formula below, note that Δ_0/E is TLS mixing angle is typically independent of TLS frequency, derivation shown in W A Phillips 1987 Rep. Prog. Phys. 50 1657).

$$\Gamma_t = \frac{\gamma^2}{2\pi \rho \hbar^4} \sum_p \frac{1}{v_p^5} \left(\frac{\Delta_0}{E} \right)^2 E^3 \coth\left(\frac{E}{2k_B T} \right),$$

where:

- γ — TLS–strain coupling constant (deformation potential), typically of order 1–10 eV.
- ρ — mass density of the host material.
- \hbar — reduced Planck’s constant.
- The sum \sum_p runs over phonon polarizations p (longitudinal and transverse).
- v_p — sound velocity for polarization p .
- Δ — asymmetry energy between the two potential wells of the TLS.
- Δ_0 — tunneling matrix element between the wells.
- $E = \sqrt{\Delta^2 + \Delta_0^2} = \hbar\omega$ — energy splitting of the TLS.
- $\coth(E/2k_B T)$ — thermal factor accounting for both spontaneous and stimulated emission/absorption of phonons

Since the frequency we perform experiment is 10-20 times lower, the TLS lifetime can be 3-4 orders of magnitude longer than that ~ 100 ns range. We are by no means trying

to make a strong quantitative argument, but at least this is quite in line with our experimental results. We revised the quoted sentence to avoid possible confusion: ‘Therefore, these results show that it is commonplace for the qubit-coupled TLS in the 100’s MHz range generally to have long lifetimes in the phononic environment of planar qubits as well.’ We have also revised a few sentences to show an open attitude for further discussion about TLS origins in the introduction “Although our study does not distinguish the microscopic origins of these TLS, ... the relaxation spectrum is well best explained by ...”, and a sentence at the end of Section E “This agreement, however, should be seen as plausible evidence rather than direct proof that our observed TLS are of dielectric origin. In addition, our observation does not rule out the existence of other completely different types of TLS that may be much weaker coupled to the qubit. On the contrary, the broad (~MHz) linewidths of the observed TLS may be seen as a hint of a denser bath of such invisible TLS, since TLS dephasing is believed to be caused by interacting with slow dynamics of other thermally-activated off-resonant TLS.”

Comment 5(a): I think that the transition rates are quite precisely determined – it follows that there are TLSs (Γ_1) that can be polarized ($\Gamma_{\delta\delta}$). There are TLSs that are long-lived which we know from measurements as shown in Fig. 2. Everything else in the analysis remains vague and does not add much to these findings. Why e.g. does the polarizability spectrum scatter so much and is essentially constant? This plot should reflect the lifetime of the TLSs.

Reply: We thank the reviewer for the questions. The main goal of our spectroscopy analysis is to have a better understanding about TLS properties in the frequency domain and provide a heuristic picture of “how much non-Markovianity/memory effect the environment has over a broad frequency range”, not necessarily to carry out precise measurements of individual TLS. While it is known in advance that long-lived TLS exist at low frequency range, we have little idea about, e.g., how much portion of TLS is long-lived, how strongly they are coupled to the qubit and whether the small average decay rates are more attributed to long live TLS or long live qubit. Applying the experiment shown in Fig 2 to do spectroscopy study is much more time-consuming as how to do a reasonable bath preparation for different coupling situations already requires some iterations to learn about the system, and data throughput is much lower due to separating bath preparation and data acquisition. In the end it remains difficult to distinguish Γ_q and Γ_t , while in our spectroscopy Γ_q , Γ_t and Γ_{qt} all have distinct features: Γ_q is the floor value of Γ_Σ spectroscopy, Γ_{qt} is the peak height in Γ_Σ spectroscopy and Γ_t is the relative height of $\Gamma_{\delta\delta}$ compared to Γ_Σ . As the reviewer also mentioned in Comment 1, one of our main findings that these relatively strong-coupled TLS are from the junction array, is related to the spectroscopy analysis. We thus respectfully disagree with the Referee's claim that ‘Everything else in the analysis remains vague and does not add much to these findings.’

For the polarizability spectrum, it is true that it scatters too much for readers to obtain useful information. The main reason is that the formula we use is true if one and only one TLS is coupled to the qubit, so when there is no prominent TLS environment it will subject to noise close to floor where both Γ_Σ and $\Gamma_{\delta\delta}$ are small, making the whole figure look scattered. Even when the formula pre-condition is satisfied, the TLS polarizability we have in experiment is further complicated by qubit initialization fidelities, so it is not fair to compare

across different TLS to judge their relative Γ_t . To better relate TLS polarizability with its lifetime, we replace the figure with a new row of data TLS polarizability (normalized, calculated by $p_k^\Delta/(Z_e - Z_g)$) as shown in Table 2.

TLS number	1	2	3	4	5	6	7	8	9	10
$g/(2\pi)$ (kHz)	50 ± 2	57 ± 3	59 ± 3	71 ± 5	75 ± 3	32 ± 5	16 ± 7	30 ± 4	66 ± 2	27 ± 2
$\Gamma_2/(2\pi)$ (MHz)	1.6 ± 0.2	1.6 ± 0.3	1.8 ± 0.2	2.0 ± 0.4	2.0 ± 0.3	5.1 ± 2.2	0.2 ± 0.2	3.5 ± 1.3	2.0 ± 0.2	2.3 ± 0.5
Frequency (MHz)	204.9 ± 0.2	220.6 ± 0.2	240.6 ± 0.3	265.5 ± 0.3	278.0 ± 0.2	298.2 ± 1.2	305.3 ± 0.2	326.3 ± 1.0	349.2 ± 0.1	360.8 ± 0.3
Γ_t (ms^{-1})	0.6 - 1.6	0.4 - 1.4	0.2 - 1.3	< 0.1	1.4 - 3.6	~ 10	< 0.1	< 0.1	1.5 - 2.6	< 0.1
TLS polarizability (normalized)	0.57	0.64	0.60	0.59	0.70	0.03	0.71	0.73	0.55	0.66

TABLE II. Extracted parameters of individual TLS in the 3D fluxonium device from Fig. 4, based on the model in Supplementary section III. Normalized TLS polarizability is calculated as $p_k^\Delta/(Z_e - Z_g)$ when it is on-resonance with the qubit, where p_k^Δ is the TLS polarization difference, approximately $\bar{Z}_H - \bar{Z}_L$ (for a more accurate expression, see Eq. (S13) in the Supplementary Materials), and $Z_e - Z_g$ can be interpreted as the maximum qubit polarization difference (determined by the reset fidelity). Their ratio is an indicator of the TLS lifetime. We note that the fitting result for TLS #7 has very large uncertainties as the peak is mostly composed of a single point, but this data point is prominent in both in back and forth sweeps, and hence is likely a discrete TLS with a narrow linewidth.

Comment 5(b): In order to extract the parameters of this convoluted protocols more precisely I suggest that the authors simulate the entire sequence (the simulation is very fast and was already implemented by the authors). For the rather low fidelity qubit reset, one has to decide for a probability distribution. Here one could use the extreme cases of a mixture of 0 and 1 or always use the average reset value. In this way one can crosscheck the current analysis, investigate to which extent demolition effects can be neglected and determine errors for the rates, in particular desired for the Γ_t rates.

Reply: The discussion of average reset value versus mixture of 0 and 1 is attached below. As the Reviewer suggested, the core question is: When qubit-TLS interact, whether assuming $P_q(0)$ is average reset value or a mixture of 0 and 1 and then average out makes a difference in average value of $P_t(t)$. Note that the following discussion assumes only one TLS but the conclusion still holds for multiple TLS interacting with qubit simultaneously.

Assume qubit is coupled to only one TLS. Let Γ_t, Γ_q be the intrinsic relaxation rates of the TLS and qubit subsystems and let Γ_{qt} be the energy exchange rate. Define the 2×2 rate matrix

$$\mathbf{R} = \begin{pmatrix} -\Gamma_t - \Gamma_{qt} & \Gamma_{qt} \\ \Gamma_{qt} & -\Gamma_q - \Gamma_{qt} \end{pmatrix}. \quad (1)$$

Let $\mathbf{P}(t) = (P_t(t), P_q(t))^\top$ and $\mathbf{b} = (\Gamma_t P_{t,\text{eq}}, \Gamma_q P_{q,\text{eq}})^\top$. For the affine system

$$\dot{\mathbf{P}}(t) = \mathbf{R} \mathbf{P}(t) + \mathbf{b}, \quad (2)$$

the solution is

$$\mathbf{P}(t) = e^{\mathbf{R}t} (\mathbf{P}(0) + \mathbf{R}^{-1}\mathbf{b}) - \mathbf{R}^{-1}\mathbf{b} = e^{\mathbf{R}t} \mathbf{P}(0) + (e^{\mathbf{R}t} - \mathbf{I})\mathbf{R}^{-1}\mathbf{b}. \quad (3)$$

Write

$$e^{\mathbf{R}t} = \begin{pmatrix} A & B \\ C & D \end{pmatrix}, \quad (e^{\mathbf{R}t} - \mathbf{I})\mathbf{R}^{-1} = \begin{pmatrix} a & b \\ c & d \end{pmatrix}.$$

Then (3) becomes

$$\begin{pmatrix} P_t(t) \\ P_q(t) \end{pmatrix} = \begin{pmatrix} A & B \\ C & D \end{pmatrix} \begin{pmatrix} P_t(0) \\ P_q(0) \end{pmatrix} + \begin{pmatrix} a & b \\ c & d \end{pmatrix} \begin{pmatrix} \Gamma_t P_{t,cq} \\ \Gamma_q P_{q,cq} \end{pmatrix}. \quad (4)$$

In particular, $P_t(t)$ is linear in the initial $P_q(0)$ and $P_t(0)$ (and likewise in $P_q(t)$), i.e.

$$P_t(t) = A P_t(0) + B P_q(0) + a \Gamma_t P_{t,cq} + b \Gamma_q P_{q,cq}.$$

$$P_q(t) = C P_t(0) + D P_q(0) + c \Gamma_t P_{t,cq} + d \Gamma_q P_{q,cq}.$$

Thus, it makes no difference for the expectation value $\langle P_t(t) \rangle$ and $\langle P_q(t) \rangle$ whether $P_q(0)$ uses average reset value $\langle P_q(0) \rangle$ or a mixture of 0 and 1.

Our conclusion is that two treatments are essentially the same since $P_t(t)$ is linear with $P_q(0)$. We have also cross-checked with a full dynamics simulation as the reviewer suggested, and don't see obvious differences in averaged value, as expected.

Comment 6: Detailed comments / recommendations:

In Fig. 1 the rates $\Gamma_{qt,s}$ should be removed: 1. they are not mentioned in the caption, 2. if these rates denote the surface TLSs, then they are already included in Γ_q , and 3. the figure looks cleaner.

Reply: We thank the Referee for the suggestion. It has been changed accordingly.

Comment 7: The paper is based on the fact that the qubit has a long lifetime Γ_q . Please state already in II. 85-89 the qubit lifetime in the absence of long-lived two-level systems. Otherwise Fig. 2 can easily be misinterpreted with Γ_{qt} being Γ_q . To get a better feeling for the device it would be good to also state the Ramsey $T2$. Can the authors be sure the echo it is not affected by the TLS memory? The reader has to find this information in the caption of Table 1.

Reply: We have added a sentence right after introduction to describe 3D device coherence property without presence of TLS :“...reaches $T_2^* = 320 \mu\text{s}$ and $T_2 = 600 \mu\text{s}$ (with Hahn echo) at half flux when not apparently impacted by near-resonant TLS”.

The $T_{2,echo}$ number is reported when in Γ_Σ spectroscopy TLS's effect is negligible at half flux. We note that both T_2 and $T_{2,echo}$ will be much worse with the presence of TLS, and it is not a low frequency noise source that can be fixed by the echo.

Comment 8: Fig. 2 caption title: “.... a bath preparation sequence.” b) ... sum of two exponential decays ... The two rates Γ_{qt} and $(\Gamma_q + \Gamma_t)/2$ appear from nowhere and are partially wrong, it should be $2\Gamma_{qt}$. You may direct the reader to the analytical formulas in Ref. 25. The rates hold true in the limit $\Gamma_{qt} > |\Gamma_q - \Gamma_t|$.

Reply: We thank the reviewer for pointing out this typo. In Supplementary section II's simulation, two rates are treated correctly. We have added the reference paper accordingly.

Comment 9: Fig. 2c): this is not the right example to illustrate a non-polarizable spot. It rather seems that there is a rather strong Γ_{qt} as well as larger Γ_t rate, since the slow rate drops to 1.4ms. Instead, one would expect a low Γ_{qt} to lead to a nearly single exponential decay that exceeds the 2ms of b).

Reply: We thank the Referee for the suggestion. We have used another set of data in Fig. 2(c) that better demonstrates a relatively non-polarizable flux point. We also note that the single exponential decay is not necessarily over 2ms since in Fig. 2(b) 2ms is the average decay of qubit and TLS, so if we assume same Γ_q for Fig. 2(b) and (c), in theory qubit intrinsic decay rate can be any value in the range of $\sim (0 - 1) \text{ms}^{-1}$.

Comment 10: The choice of using Z and p is not very wise and not consistently used throughout the paper. In the appendix p is also used to indicate qubit populations. I do not see any benefit of using polarizations. I would suggest to stick to the notations that were used in Refs. 20, 22 and 25. This simplifies the comparison, in particular since many formulas are rederived here. It does not hurt to add further citations and would help the reader to find the original derivations, especially in the appendix.

Reply: We thank the reviewer for pointing out the inconsistency. We use Z to represent qubit to emphasize that all “populations” are in -1 to 1 scale instead of 0 to 1 . At the end we decide to make Z consistently represent qubit in the supplementary. We have added those citations to Supplementary.

Comment 11: Fig. 3: The rather large t_1, \dots, t_4 values are somewhat in conflict with $1/\Gamma_{qt} \sim 20\mu\text{s}$ of Fig. 2. As the authors also explain in the main text, the 4-point exponential fit only makes sense when the bath polarization changes slowly (many TLSs), otherwise t_2 measures a drastically altered TLS population from the previous polarization at t_1 . However, why should we use this sequence in the first place and not always the CD-8 that the authors also mention, especially since the authors do often resolve individual TLSs? The CD-8 gives the same data throughput and is less prone to exponential fit misinterpretations.

Reply: Indeed, the FD-4 protocol is not designed to extract very fast decay rates, and this is precisely why we introduced the CD-8 protocol for situations where $1/(2\Gamma_{qt}) \sim 20\mu\text{s}$.

Our choice to prioritize FD-4 as the main spectroscopy sweep protocol is motivated by three practical considerations:

- (1) **Reliable background decay extraction in the absence of strong TLS features.**
In many frequency regions where no prominent TLS is present and the bath dynamics are slow. We would like to have an efficient assessment of how Markovian this background decay is. In these cases, FD-4 captures the global shape of the decay curve more reliably for single-exponential fitting. Such background data are essential for extracting Γ_q , and FD-4 provides higher contrast and higher signal-to-noise under these conditions. While our work is probably going to be more recognized for the discrete TLS spectrum, measuring the background Γ_q accurately in the presence of the forest of TLS Lorentzians is just as important to us.
- (2) **Robustness when the decay rates are small.**
If CD-8 is applied first in a regime where decay is slow, the limited contrast over the fixed $0\text{--}70\mu\text{s}$ range can make it difficult to detect whether the protocol is failing to extract the correct rate. With FD-4, if the decay rate happens to be large, the protocol itself naturally signals this (e.g., a fitted rate $>4/\text{ms}$ in our study), prompting us to switch to CD-8 or other specialized protocols. This ordering minimizes the risk of misinterpreting weak-contrast CD-8 data.
- (3) **Efficiency of data collection.**
CD-8 has inherently lower data throughput: half of the sequences are used solely to equalize the interaction duty cycle and do not produce new measurement data. FD-4

is therefore more efficient for a full spectral sweep, especially when many frequency points exhibit weak TLS coupling.

We emphasize that CD-8 is introduced as a targeted tool for strong qubit–TLS coupling, where the bath evolves significantly within the typical delay-time window. For these specific frequency regions, CD-8 indeed provides clearer interpretation and avoids exponential-fit ambiguities. However, using CD-8 exclusively across the entire spectrum would be significantly slower without offering advantages in the weak-coupling regions.

Finally, we note that the overall approach— T_1 -type measurements with controlled sequence history—is quite flexible. One could, in principle, extend CD-8 with more and longer delay points so that a single protocol works for both fast and slow bath dynamics. Such a unified CD-8 protocol is possible, but it would lower data throughput because many delay points would not be informative at a given frequency. This reflects an inherent trade-off between universality and efficiency. Future studies may adopt a unified CD-8-like protocol with more delay points to cover all regimes, or dynamically adapt the protocol based on online estimates of the bath polarization rate. Here, our aim is simply to present FD-4 as an efficient general sweep and CD-8 as a targeted tool for strong qubit–TLS coupling, rather than to suggest that one protocol should replace the other universally.

To make the point clearer to readers, we added a sentence in main text section D:

“This spectroscopy method is intended and most effective as a substitute of traditional qubit T_1 spectroscopy to unwind possible memory effects from slow-varying weakly-coupled environment baths. On the other hand, it can also be used to investigate discrete TLS that dominates the qubit dynamics, using shortened measurement blocks with cycling delays (see methods).”

Comment 12: My suggestion would be to simplify Fig. 3. In a), explain the constant repetition time sequence CD-8 of the appendix and then show c) and d). Explain in the appendix the version with the different timings and why these protocols are also needed/beneficial. Please use not too many different notations for the same thing, I would suggest that you stick to the transition rates $\Gamma_{up,down}(0)$. It would be very helpful to show the estimated TLS population versus time T.

Reply: As we discussed in Comment 11, FD-4 is the fundamental protocol and CD-8 is designed for large Γ_{qt} case, so we would like to start from the fundamental one.

On the notation side, we transfer Γ_{\uparrow} and Γ_{\downarrow} into Γ_{Σ} and Γ_{δ} due to spectroscopy analysis requirements. These two decay rates are further related to TLS state H and L, which is also necessary even if we use Γ_{\uparrow} and Γ_{\downarrow} as notation.

Since we lack e.g. TLS equilibrium population for full dynamics simulation to demonstrate estimated TLS population at that point, it would be confusing to display the TLS population here in Fig 3. We do have simulated qubit/TLS dynamics shown in Supplementary section III (D).

Comment 13: Section C beginning: The inverse of $T1$ is commonly Γ_1 , there is no need to define something new. The authors also use Γ_2 . The time-dependence (T) of the rates is missing. It is unclear e.g. in Fig.4a) at which time the rates are taken. In order to judge whether a comparison of the Γ_1 rates makes sense to exclude a bosonic environment the reader must know the TLS polarization that is reached. In Fig. 4a) the points scatter several sigma from the red line, which means in principle that in some places there is a bosonic environment, however, since the points scatter in both directions, it actually means that the analysis of the data is prone to errors or that the statistical errors are not calculated correctly. The authors simply show that the rates are correlated, which is not at all surprising. Instead, one has to show at each flux position that the rates are identical, for this the authors may use the rates as a function of the time T . In the current form, all statements derived from this plot are not supported.

Reply: We thank the Referee for the detailed comments. In the paper we have stated that T_1 is not a well defined concept under a non-Markovian environment. Though Γ_Σ is known as $1/T_1$ under Markovian assumption, under TLS environment influence one needs to write $\Gamma_\Sigma = \Gamma_q + \Gamma_{qt}$ and consider each contribution instead of using a simple exponential fit to extract Γ_1 . Γ_Σ , $\Gamma_{\delta,H}$ and $\Gamma_{\delta,L}$ ($\Gamma_{\delta\delta} = \Gamma_{\delta,L} - \Gamma_{\delta,H}$) are all time-independent rates. H and L represent the highest and lowest TLS polarizations we can reach by our protocol, respectively. To more intuitively understand those rates, in Fig. 3(b) lower panel, (c) or (d), if we assume perfect qubit initialization, then first point of red curve roughly represents $-2\Gamma_{\downarrow,L}$, while last point is roughly $-2\Gamma_{\downarrow,H}$. On the other hand, the first and last points of the blue curve are $-2\Gamma_{\uparrow,H}$ and $-2\Gamma_{\uparrow,L}$, respectively.

As for Fig 4(a), we have identified a deficiency in previous analysis of the uncertainties (which relied on the uncertainty from curve fittings, but underestimated the uncertainty when the data points coincidentally match the model nearly perfectly), and we have corrected that problem. We have also added more data to the plot as well. Note that each point represents a single flux point and if a bosonic environment that has a long memory effect exists there, it can be polarized to some degree and thus let $\Gamma_{\Sigma,H} > \Gamma_{\Sigma,L}$. The fact that our linear fit slope is close to 1 indicates an overall rather small contribution from the bosonic bath environment, so that we assume $\Gamma_{\Sigma,H} = \Gamma_{\Sigma,L}$ in later analysis. Since indeed the slope is slightly less than 1, some small bosonic memory effect may indeed be present, so we have weakened our statements with respect to the absence of long-live bosonic baths at multiple places in the paper due to our data quality limitation.

Comment 14: The rate $\Gamma\delta$ and $\Gamma\delta\delta$ are time-dependent. It is not clear at which time they are evaluated. It is not explained how the \bar{Z} average is computed exactly.

Reply: $\Gamma_{\delta\delta}$ is indeed time-dependent and we did not record the value. As for Γ_{δ} , it is time-independent and relates to all decay rates and experiment protocol, as it evaluates $(\Gamma_{\downarrow} - \Gamma_{\uparrow})$ for TLS in H state versus in L state.

We have added explanation to initial slope extraction and \bar{Z} calculations in Supplementary section III ‘Modeling two-timescale relaxometry’, as shown below in the screenshots.

Assume we can initialize our qubit towards two different states, “excited” and “ground”, with $Z = Z_e$ and $Z = Z_g$, respectively. Qubit state preparation is not ideal in practice, but in general, $Z_e > 0$ and $Z_g < 0$. We prepare each state, apply different delay times t_i , and perform a single-exponential fit to the measured readout populations Z_i to extract the initial decay slope:

$$Z = b_g - \left(\frac{dZ}{dt}\right)_0 \tau e^{-t/\tau}, \quad (\text{S5})$$

where b_g and τ are the background and time constant of the exponential fit, respectively.

Now, we have two equations to solve two unknowns ($\Gamma_{\Sigma}, \Gamma_{\delta}$):

$$\begin{aligned} \left(\frac{dZ}{dt}\right)_{0,e} &= -Z_e \Gamma_{\Sigma} - \Gamma_{\delta}, \\ \left(\frac{dZ}{dt}\right)_{0,g} &= -Z_g \Gamma_{\Sigma} - \Gamma_{\delta}. \end{aligned} \quad (\text{S6})$$

.....

In our experiment, time-averaged qubit polarization is expressed as:

$$\bar{Z} = \frac{\sum_i \left[\int_0^{t_i} b_g - \left(\frac{dZ}{dt}\right)_0 \tau e^{-t/\tau} dt + Z_{g/e} t_{\text{idle}} \right]}{\sum_i [t_i + t_{\text{idle}}]}. \quad (\text{S14})$$

$Z_{g/e}$ is determined by qubit initialization condition (details in Fig. S3 and Section IV), and t_{idle} is the idle time after pumping (15 μs here, and details in Table S1). In the 3D device, $t_i = 1, 40, 150, 330 \mu\text{s}$ for FD-4 and $t_i = 1, 10, 20 \dots 70 \mu\text{s}$ for CD-8. By substituting the corresponding values of b_g , τ , and $\left(\frac{dZ}{dt}\right)_0$ from the final bath-polarizing cycles into the above expression, one obtains \bar{Z}_H and \bar{Z}_L .

Comment 15: II. 214 formula holds true only for 1 TLS.

Reply: This assumption was mentioned in Extended Data Fig. 4 caption in the last edition, and now we added this requirement in the main text.

If there is only one prominent TLS, the fraction, $\frac{\Gamma_{\delta\delta}}{2(\Gamma_{\Sigma} - \Gamma_q)}$, reflects the TLS population difference between its high and low states.

Comment 16: Eq. 4: Z needs the index k.

Reply: Z does not need index k since it represents the only qubit, not k^{th} TLS.

Comment 17: It is not explained how Γ_t is extracted. In the methods section the parameter τ_e is introduced (valid for 1 TLS). τ_e should approximately be the inverse of $\eta\Gamma_{qt} + \Gamma_t$. Especially for such a delicate parameter like Γ_t it would be important to get a range of uncertainty.

Reply: Indeed τ_e can be interpreted as inverse of $\eta\Gamma_{qt} + \Gamma_t$ if there is always only 1 TLS affecting the qubit. While the simple single exponential model works well for us to extract Γ_{Σ} , $\Gamma_{\delta,H}$ and $\Gamma_{\delta,L}$ rates, it is difficult to extract reliable and self-consistent Γ_t information from τ_e , which is potentially caused by multiple TLS with weaker interaction (Γ_{qt}) but long intrinsic lifetime, or due to a relatively long time period between two neighboring sequence blocks (1.6ms for FD-4 and 0.4ms for CD-8). Thus we did not try to extract $\Gamma_{t,k}$ from individual flux point measurements but instead a global spectroscopy fit for $\Gamma_{\delta\delta}$. We have added a sentence below in the supplementary to explicitly point out how Γ_t is extracted:

“For both protocols, we first extract all $\Gamma_{qt,k}$ -related parameters (g_k , $\Gamma_{2,k}$ and $\omega_{t,k}$) and Γ_q from the Γ_{Σ} spectrum, and then do the $\Gamma_{\delta\delta}$ spectroscopy analysis to obtain $\Gamma_{t,k}$ information.”

Comment 18: II.71 (on the order of ms^{-1}) What does it refer to? Coupling strength? Qubit loss?

• Subscript labels should not be in italic. In the supplement it is sometimes done correctly.

Reply: The much weaker continuum of surface dielectric loss (in spectroscopy analysis called qubit background decay rate Γ_q) is mainly due to dielectric surface loss of substrate-air interface, metal-substrate interface excluding junction chain region and metal-air interface.

We have changed subscript labels format in the paper so that the label is Italic only when it represents an index.

Comment 19: For consistency: $\Gamma_{2,k}$ with the comma

Reply: We have added commas for $\Gamma_{2,k}$, and for consistency we have also added commas in multiple symbols.

Comment 20: II. 31 Grünhaupt et al. Nature Materials 2019 would be a better fit.

Reply: We thank the Referee for the suggestion. We have added this citation there.

Comment 21: Eq. S6 needs a citation. How is the total dephasing rate defined?

Reply: We have added a citation and total decoherence rate definition here.

where g_k is the coupling rate between qubit and k^{th} TLS, and $\Gamma_{2,k} = \frac{\Gamma_q}{2} + \frac{\Gamma_{t,k}}{2} + \Gamma_{\phi,q} + \Gamma_{\phi,k}$ is the total decoherence rate of the qubit and k^{th} TLS (in our case $\Gamma_q, \Gamma_{t,k}, \Gamma_{\phi,q} \ll \Gamma_{\phi,k}$).

Comment 22: Eq. S30, the matrix upside down.

Reply: We thank the Referee for the careful check. The matrix is indeed opposite and we have corrected the mistake.

$$[\Gamma] = \begin{bmatrix} -\Gamma_q - \Gamma_{qt} & \Gamma_{qt} \\ \Gamma_{qt} & -\Gamma_t - \Gamma_{qt} \end{bmatrix}$$

Comment 23: Fig. 1 needs scale bars.

Reply: Fig. 1(a) and (b) are both just cartoons, so the qubit is not drawn strictly to scale and the Josephson junction array part is intentionally magnified. It is difficult to present the full qubit geometry to scale. Thus we cannot put scale bars here.

Comment 24: Fig. S4 colorbar label is missing

Reply: The colorbar label has been added to the figure. We used another set of data with calibrations to qubit σ_z instead of phase contrast.

• I think at this point it is clear that the manuscript has to be thoroughly revised, such that the overall quality and transparency is improved.

Reply: We thank the Referee for all the suggestions and questions, and we believe that they have all been addressed properly so that overall quality has been improved to better satisfy publication requirements of Nature Communication.

Reviewer #2 (Remarks to the Author):

Comment 1: The authors propose and experimentally demonstrate a technique for studying two-level systems that are weakly coupled to a qubit, as well as their effect on qubit relaxation. A key feature of the method is that the environment initialization is performed simultaneously with the qubit relaxation measurement, which significantly reduces the total measurement time compared to conventional approaches.

The manuscript is well written, and I believe the proposed relaxometry approach will be useful for studying noise sources, improving device characterization, and developing more coherent qubits. However, I recommend the authors address the following points.

Reply: We are grateful to the Referee for an excellent summary and overall positive evaluation of our work.

Comment 2: Major points:

The authors mention that, in conventional relaxation measurements, the results can depend on the choice of time step, reset protocol, and averaging techniques due to non-Markovian effects. I believe a similar analysis should also be provided for the data obtained using the proposed technique. In particular, it would be helpful to clarify how the choice of time step and time range affects the extracted quantities, especially considering that the derivative is determined from an exponential fit to $Z(t)$, while the actual Z dependence on time differs from a purely exponential form.

Reply: We thank the reviewer for the suggestion that a full dynamics simulation would make our proposed protocol more convincing and a more detailed explanation about time step choice is necessary for people who would like to adopt this method. We have added a subsection as Supplementary III(E) (as shown below) to verify our method according to datasets generated by full dynamics simulations.

E. Verification of methods via full dynamics simulation

Given the greater complexity of our protocols compared to a conventional T_1 measurement, we performed a self-consistency check to verify that our protocol and analysis can reliably extract the underlying qubit-TLS system parameters. Assuming that the qubit interacts with a single TLS, we specify hypothetical sets of system parameters— Γ_q , Γ_t , the qubit-TLS coupling strength g , and the total decoherence rate Γ_2 , among others—and use full dynamical simulations of our experimental protocol to generate synthetic measurement data, including noise levels comparable to those in Fig. S2. We then apply our analysis procedure to this simulated data and compare the extracted parameters with the originally assigned ones.

In this simulation, we examine two representative cases for our 3D chip: the qubit coupled to a TLS with a millisecond-scale lifetime (2 ms) and to one with a sub-millisecond lifetime (200 μ s). To further test the generality of our protocol for devices with larger Γ_q (such as our planar device), we vary Γ_q from 0.5 ms^{-1} to 5 ms^{-1} . All other parameters are kept the same across the parameter sets: both Z^{eq} and p_k^{eq} are set to -0.1

(corresponding to 45% thermal population). The qubit is always pumped toward $|e\rangle$, with $Z_e = 0.5$ and $Z_g = -0.5$. The TLS frequency is fixed at zero detuning, and the qubit frequency is swept from -10 MHz to +10 MHz in 1 MHz steps. For simplicity, assigned $\Gamma_{qt} > 4 \text{ ms}^{-1}$ would trigger CD-8 protocol, otherwise the data is generated by FD-4 protocol.

The spectroscopy analysis results are shown in Fig. S4 and Table. S2. We find that Γ_t tends to be slightly overestimated, primarily due to the non-negligible idle time following the reset pulse when the qubit-TLS dynamics is fast. Nevertheless, the extracted parameters show good overall agreement with the nominal values. We therefore conclude that, despite minor deviations arising from the quasi-steady-state approximation, our model reproduces the principal behavior of the qubit-TLS coupling dynamics.

FIG. S4. **Spectroscopy analysis from the full dynamics simulation data.** Γ_Σ and $\Gamma_{\delta\delta}$ decay rates extracted from the full dynamical simulations for different combinations of Γ_q and Γ_t , with same-colored curves showing their corresponding fits.

		Γ_q (ms ⁻¹)	Γ_t (ms ⁻¹)	f_{TLS} (MHz)	$\frac{g}{2\pi}$ (kHz)	$\frac{\Gamma_2}{2\pi}$ (MHz)
Parameter set 1	Assigned	0.5	0.5	0	50	2
	Extracted	0.55 ± 0.22	0.86 ± 0.33	0.05 ± 0.08	47 ± 2	1.6 ± 0.2
Parameter set 2	Assigned	0.5	5.0	0	50	2
	Extracted	0.45 ± 0.17	5.5 ± 1.1	0.01 ± 0.07	50 ± 1	2.0 ± 0.1
Parameter set 3	Assigned	5.0	0.5	0	50	2
	Extracted	5.0 ± 0.2	0.81 ± 0.26	0.06 ± 0.06	48 ± 1	2.1 ± 0.1
Parameter set 4	Assigned	5.0	5.0	0	50	2
	Extracted	5.4 ± 0.1	7.4 ± 1.7	0.10 ± 0.06	44 ± 1	1.9 ± 0.1

TABLE S2. **Assigned and extracted parameters from full dynamics simulation** We consider four parameter sets that vary Γ_q and Γ_t , using nominal intrinsic lifetimes of 2 ms and 200 μ s as references.

In terms of the choice of time step and time range, we revised a sentence in main text section B to explain more explicitly how we choose these delay times “We note that a four-point exponential fit can usually yield \dot{Z}_0 robustly over a broad range of qubit relaxation timescale, from $\sim t_1$ to 5-10 times of t_3 , which motivates our choices of t here.” We also added more description on how to choose time step and time range in Supplementary section III (D).

Specifically, our primary goal is to extract the initial slope $(\frac{dZ}{dt})_0$ (Eq. (S5)) reliably from the measured data at various delay times, while the accuracy of b_g and τ is of less concern. In practice, the shortest probe time t_1 is chosen based on the fastest expected decay timescale ($\sim 1/(2\Gamma_{qt})$), whereas the longest probe time t_3 is selected to be long enough to capture a reasonable decay amount, for example, about 20% of the full decay contrast, under the slowest decay dynamics. We note that the long environmental response time we often observe is mainly contributed by some less-coupled but polarizable TLS such as those in the junction chain but farther-detuned, so in the spectroscopy sweep we choose a measurement time range ($T_{\text{max}} \approx 20$ ms) up to a few times of expected longest TLS lifetime we can identify.

Comment 3: The authors should be more careful in citing prior work (lines 15–16). The phrase “studied in relatively contrived demonstrations” seems inappropriate, as non-Markovian noise has been investigated in real devices. For example, studies on quasiparticles [S. Gustavsson et al., Science 354, 1573–1577 (2016)] and near-resonant TLS and TLFs [J. Burnett et al., npj Quantum Information 5, 54 (2019)] have explored non-Markovian effects in realistic experimental settings. The statement in lines 219–222 is also inaccurate in light of [J. Lisenfeld et al., Nat. Commun. 6, 6182 (2015)] and other related

works. I recommend the authors revise these formulations and include appropriate references.

Reply: We sincerely thank the Referee for pointing out the relevancy of these studies. In response, we have removed the phrase “studied in relatively contrived demonstrations (i.e. where the environment is engineered to be non-Markovian)”. As now clarified in the introduction, we explicitly acknowledge previous investigations of non-Markovian effects arising from quasiparticles and spurious two-level systems, which can exhibit long-time memory in their interaction with the qubit, as well as related studies of correlated dephasing noise. We also emphasize that, despite these notable examples, “there have been relatively limited attention to non-Markovian effects in common practices of characterizing qubit relaxation dynamics”, which is the specific context of this work. To further improve the discussion, we have added the two suggested references to better situate our study within the broader literature.

The statement in lines 219–222 has now been revised to explicitly exclude the scenarios “except when coherent qubit-TLS oscillation is explicitly observed and modeled, e.g. [33]” to avoid confusion, where we also explicitly cite Lisenfeld et al. (Nat. Commun. 6, 6182 (2015)), which observes coherent dynamics in the strong-coupling regime among other studies. Our revised formulation emphasizes that our goal is to determine both self- and cross-relaxation rates in the weak-coupling, relaxation-dominated regime, which is not accessible in those strong-coupling studies.

Comment 4: Minor points:

It may be helpful to indicate the corresponding peaks from Extended Data Table 2 directly in Figure 4(c), for example with numbers or vertical dashed lines, as some are difficult to distinguish. This would improve readability.

Reply: Now we have updated Fig 4(c) with vertical dashed lines showing where TLS are located. Numbers correspond to serials in Table 2.

For consistency, we did the same revision for the planar device spectroscopy in “Methods” as well.

Comment 5: In lines 198–199, the authors argue that the observed similarity between the relaxation rates in the low and high environment states ($\Gamma_{\Sigma L} \approx \Gamma_{\Sigma H}$) indicates the absence of non-Markovian effects from a bosonic bath. However, this conclusion is not entirely evident. Could the authors clarify how this similarity excludes non-Markovian contributions from bosonic environments? A more detailed justification would strengthen the argument.

Reply: If qubit is coupled to bosonic environments, e.g. cavity or some multi-level defects, heating up the environment corresponds to increasing occupancy numbers, which can lead to Γ_{Σ} rising. Specifically, qubit coupled to a bosonic mode leads to Γ_{Σ} in proportional to $(2\bar{n} + 1)$ (\bar{n} is the average occupancy number in the bosonic mode), which is not observed in our experiment. We have revised the sentence to make it clear. We have also weakened our statement considering the relatively large error bars.

We find that the measured results are mostly consistent with $\Gamma_{\Sigma L} \approx \Gamma_{\Sigma H}$ (Fig. 4(a)), suggesting that there is relatively little contribution to the environment memory effect from possible bosonic baths, which would have increased Γ_{Σ} by a factor of $2\bar{n} + 1$ (\bar{n} is the average occupancy number in the bosonic mode).

Comment 6: The reset fidelity in the 3D architecture shows significant variation. Could the authors explain why this reset method was chosen over the one used in the planar device, particularly since the proposed relaxometry relies on efficient initialization?

Reply: We acknowledge that if fast and high fidelity initialization can be done, it will make the experiment process and data analysis easier and cleaner. Applying a non-QND readout pulse to reset the qubit involves qubit and cavity’s higher energy states’ excitation exchange, a mechanism that is not fully understood yet, and thus larger reset fidelity variation across frequency is not surprising. We would like to have a similar reset strategy based on FPGA feedback (active reset) on the 3D device if possible, but two practical limitations prevent us from doing so:

- (1) For the 3D device, $\chi \ll \kappa$, so that the device parameter is not ideal to do high-fidelity active reset.
- (2) Our measurement setup for the 3D device did not contain FPGA electronics capable of fast feedback operations.

Alternatively, there is another common method resetting fluxonium qubit that involves sideband cooling (W. Wang *et al.*, PRL 2024), which we find more difficult to carry out (than the non-QND readout-based method) on our device for a wide frequency range.

Comment 7: The T_2 and T_{2E} values were measured when TLS did not strongly affect the qubit. Could the authors clarify how this condition was ensured? Was it achieved by detuning the TLS from the qubit or by excluding the data points where strong coupling was observed?

Reply: Coherence is measured at half-flux and we don't have the ability to manipulate TLS frequency. In a spectroscopy experiment when we see close-to-floor Gamma rates (Fig. 4(c)) at half-flux, we would do T_2 and $T_{2,echo}$ measurement and record the values. The treatment is essentially to filter out the data points when the relaxation rate is notably higher than the background rate.

Comment 8: Supplementary, line 108: For improved readability, I suggest replacing the reference to Eq. (S23) with a reference to Eq. (S5).

Reply: We thank the Referee for noticing this problem. We have fixed this issue according to the referee's suggestion.

Comment 9: Supplementary, lines 177–178: The authors attribute the observed readout phase difference at low pumping to residual reset effects from the previous pulse. Why does this effect appear only at frequencies above 380 MHz?

Reply: As mentioned in the reply to comment 6, this reset scheme depends on resonator and fluxonium qubit frequency placement, and likely other mechanisms including the involvement of off-resonant TLS, which hasn't been systematically understood, so we cannot predict optimized pumping amplitude choice. However, our readout tone amplitude is quantitatively comparable to our pumping tone amplitude choices, so it may not be surprising that in certain frequency range the readout tone can already sufficiently reset fluxonium qubit to $|g\rangle$ state.

Comment 10: Figure S8 caption: There appears to be an inconsistency – '1' and '0' should likely be replaced by '↑' and '↓'. Also, the phrase "pumping and initialization are in the same direction" may be misleading. A clearer version could be: "The '↑' initialization requires only pumping, while the '↓' initialization requires pumping followed by an additional π pulse."

Reply: We thank the Referee for the careful attention. It has been adjusted accordingly.

Comment 11: I believe the manuscript presents a valuable experimental approach with potential impact on noise characterization. Provided the authors address the points raised above, I would support the publication of this manuscript.

Reply: We thank the reviewer for supporting this manuscript being published, and all the insightful comments which helps improve the readability of the manuscript.

Reviewer #3 (Remarks to the Author):

Comment 1: The authors explore the relaxation of a fluxonium qubit and a 3D cavity and find that it does not fit the usual Markovian assumptions with simple T1/T2 times. This is shown to be due to coupling to a TLS bath, and the dynamics of interaction and joint relaxation of the fluxonium qubit and the TLSs are explored. The authors introduce a new protocol to efficiently study this process, where the qubit relaxation is probed during a long polarization of the TLS bath to the excited and ground states. The authors show quantitative agreement with a simple model for a qubit coupled to several TLS defects, and are able to extract relevant rates for the decay and interaction of the systems. Remarkably, by varying the fluxonium frequency in the low frequency regime 0.1-0.4 GHz, the authors observe a significant change in the relaxation dynamics - and show they can be related to several resonant TLS defects coming to resonance with the qubit in that frequency regime. The manuscript is clearly written and well explained. We believe the manuscript clarifies a mystery behind the lifetimes of fluxonium qubits and presents a new way to quantify the TLS bath coupled to it. Thus, it is relevant to the general community studying the coherence of superconducting circuits, and the more fundamental study of interaction with a spin bath in general. We believe it is suitable for publication in Nature Communications.

Reply: We thank the referee for their positive assessment of our work. We especially appreciate the referee's evaluations highlighting the importance of our study and support for the publication.

Comment 2: We have several questions that arose while reading the manuscript, which can hopefully be better explained:

In fig. 3d, the authors mention that the fast relaxation is not due to the intrinsic TLS lifetime being shorter, but rather due to stronger interaction with the qubit. This can also be seen in the initial rates in Fig. 3d being significantly higher than those of Fig. 3c. In that case, is the TLS relaxation rate just given by the background decay rate of the fluxonium qubit (without TLS) or is it a more complex interaction?

Reply: In the case of Fig. 3d, the TLS decays through the qubit, but its apparent decay rate is mostly *not* set by the background decay rate of the fluxonium qubit, nor is it strongly influenced by the experimental protocol. The experimental protocol frequently resets the qubit to $|g\rangle$ or $|e\rangle$, which, under strong qubit-TLS interaction, effectively provides a strong decay channel for the TLS on a faster time scale than either Γ_q or Γ_t . We have added a few words in parenthesis to suggest the role of qubit reset rather than its background decay rate in this dynamics:

“... this fast bath dynamics is often due to strong relaxation through the qubit (which receives frequent reset) and should not be taken as a direct measure of the intrinsic TLS lifetime.”

Comment 3: Can these measurements also be used to extract the intrinsic TLS lifetime (due to e.g. phonons) reliably? This can be particularly interesting in systems such as Ref. 22 where the phonon bath can be tuned.

Reply: According to our evaluation, our protocol can extract TLS lifetime with a reasonable agreement but not precisely. We have also added a subsection titled “Verification of methods via full dynamics simulation” in Supplementary Section III(E) that provides a more quantitative idea about the reliability, and note that one could get better agreement with more efficient reset. We also note that our choice of delay times serve the purpose of wide range spectroscopy study. If one would like to obtain certain TLS information more accurately, those delay time choices can be more specified to improve applicability to different TLS, or even use other measurement protocols, e.g. a measurement similar to Fig. 2 after knowing the qubit background decay rate Γ_q if TLS lifetime is similar or longer than the qubit.

Comment 4: The observation that the decay rates at positive and negative polarization are equal ($\Gamma_{\Sigma H} \sim \Gamma_{\Sigma L}$) implies that the temperature of the TLS bath does not affect the dynamics. However, the authors also comment that it indicates “an absence of non-Markovian effect from any bosonic bath”. This statement should be clarified. Which bosonic bath is assumed and why is this seen as an absence of it. In general, is it clear that these defects which interact with the qubit have only two levels and not more?

Reply: If qubit is coupled to bosonic environments, e.g. cavity or some multi-level defects, heating up the environment corresponds to increasing occupancy numbers, which can lead to Γ_{Σ} rising. In specific, qubit coupled to a bosonic mode leads to Γ_{Σ} in proportional to $(2\bar{n}+1)$ (\bar{n} is the average occupancy number in the bosonic mode), which is not observed in our experiment. We have revised the sentence to make it clear.

We find that the measured results are mostly consistent with $\Gamma_{\Sigma L} \approx \Gamma_{\Sigma H}$ (Fig. 4(a)), suggesting that there is relatively little contribution to the environment memory effect from possible bosonic baths, which would have increased Γ_{Σ} by a factor of $2\bar{n} + 1$ (\bar{n} is the average occupancy number in the bosonic mode).

Comment 5: This manuscript focuses on defects in the AlOx layer as the main source of TLS, while previous work such as Ref. 20 discussed them more generally without being certain of their origin. Specifically, magnetic defects were also considered as options. Would this also be a possibility in this experiment or can it be excluded? How would the authors explain the change of the TLS spectrum with time (Fig. 4b), which perhaps seems more reasonably due to magnetic defects?

Reply: Thanks to the Referee for noticing the possibility of magnetic defects. While we cannot rule out the possibility of magnetic origin, we believe that the TLS we observed in the spectroscopy are mainly defects in the AlOx layer because of two reasons:

- (1) Based on research on two-level systems today (like S. Schlör et al., PRL 2019), TLS with charge coupling are also regarded as unstable and their properties can fluctuate with time. Our spectroscopy fitting model includes the hypothesis that TLS originated from electron-spin, and parameters extracted are self-consistent.

(2) If we assume features we see in the spectroscopy are due to magnetic defects couple via its magnetic dipole μ to the qubit's oscillating magnetic field B , the coupling is given by $g \sim \mu B / \hbar$. We estimate the corresponding oscillating current amplitude as $2ef_{01} \langle 0|n|1 \rangle$, which yields approximately 25pA, leading to a maximum magnetic field in the small junction on the order of 0.1nT. To match the coupling g we observed (tens of kHz), one would probably require a large spin cluster (on the order of thousands of μ_B), which is not common in previous reports for a naturally occurring, unengineered environment.

Therefore, while we cannot rule out magnetic defects a priori, the coupling magnitude strongly favors dielectric TLS in the AlOx barrier as the dominant mechanism rather than magnetic origin. Nevertheless, we agree that coupling strength only is not a strong enough proof as TLS origin remains mysterious. We therefore added a sentence in the introduction "Although our study does not distinguish the microscopic origins of these TLS, ... the relaxation spectrum is well best explained by ...", and a sentence at the end of Section E "This agreement, however, should be seen as plausible evidence rather than direct proof that our observed TLS are of dielectric origin. In addition, our observation does not rule out the existence of other completely different types of TLS that may be much weaker coupled to the qubit. On the contrary, the strong dephasing of the observed TLS may be a hint of a denser bath of such invisible TLS."

Comment 6: In the model that included Γ_{σ} and Γ_{δ} (Eq. 3,4), a duty-cycle factor was introduced because the interaction was not on for the entire duration. Why does this only affect the Γ_{δ} term and not the general decay term Γ_{σ} ?

Reply: For a detailed derivation of Γ_{Σ} and $\Gamma_{\delta\delta}$ formulas, we have included them in supplementary section III. Here we can provide a more intuitive explanation. It is because only $\Gamma_{\delta\delta}$ is sensitive to the fraction of time during which the qubit-TLS interaction is "active" as it will vary depending on the "H" and "L" states TLS polarization difference. This fraction is exactly what the duty-cycle factor encodes. In contrast, Γ_{Σ} represents the total decay rate of the qubit (including qubit intrinsic decay rate and qubit-TLS energy exchange rate) and does not depend on how effectively the protocol polarizes the TLS. For this reason, the duty-cycle factor modifies $\Gamma_{\delta\delta}$ but has no effect on Γ_{Σ} .

Reviewer #1 (Remarks to the Author):

Comment 1: This revised manuscript is in my view a significant improvement over the previous version. The authors have addressed all the questions in my original review. I therefore recommend that the manuscript is published in Nature Communication.

Reply: We thank the reviewer again for the questions raised in the previous review, and the support for publication in Nature Communications here.

However, I would like to insist on two minor changes:

i) Regarding the new plots Fig. 2b, c, please set the y-axis on the same range. In this way one can see more clearly that the non-polarized T_1 starts at $\sigma_z=0.4$, which coincidentally is the point of inflection in panel b. If the data can be compared to Fig. 4 (measured roughly at the same time), it would be very valuable to add the qubit frequencies to panel b, c.

Reply: We have updated Fig. 2b and c that set the y-axis on the same range. Regarding qubit frequencies, data in Fig 2b and c is taken in a different time from spectroscopy sweep in Fig 4's. But it is also a good idea to show two plots' data were taken at different flux points in the plot.

ii) The authors forgot to update the citation (s. comment 20).

Reply: We have updated the citation in this revision.

FYI reply to comment 5b:

I have realized that my remark on the qubit reset distribution was premature and conflated with the issue of state-preparation errors in the active reset. However, in your reply to comment 5b, you implicitly assume that the qubit reset distribution is independent on the

qubit population before the reset. Any such dependence can be interpreted as a state preparation error in your protocol. However, this error seems to be small in your experiment, as can, to some extent, be verified from Fig. S8.

Reply: We thank the reviewer for the information.

Reviewer #2 (Remarks to the Author):

The authors have carefully addressed all of my comments and questions. The revised manuscript has been significantly improved, and the clarifications and additional explanations resolve my previous concerns. I find the responses to be clear and satisfactory. I recommend the manuscript for publication.

Reply: We thank the reviewer for careful evaluation of the manuscript, and for the support of the publication of this work.

Reviewer #3 (Remarks to the Author):

The authors have made significant corrections and improvements in the revised version, and I believe they have addressed the questions and issues raised by myself as well as the additional reviewers. I do not have any further comments, and happily recommend this manuscript for publication in Nat. Comm.

Reply: We thank the reviewer for constructive input and support for publication.

Reviewer #4 (Remarks to the Author):

Reply: We thank the reviewer for the effort reviewing this manuscript.

In this work, the authors prove the existence of long-lived two-level systems in state-of-the-art Josephson junction array (JJA) fluxoniums. These (most likely the same) long-lived TLSs were first discovered and characterized in granular aluminum. However, many questions remained open, especially regarding the nature of these TLSs, to which extent they are specific to the disordered nature of granular aluminum, and if these TLSs are also relevant for junction array fluxonium and other superconducting qubits.

While, in retrospect, there are indications of their presence in earlier JJA fluxonium experiments, as the authors also state, the results presented in this manuscript as well as the results of the Munich group presented at the March meeting, are the first and eagerly awaited scientific documentations of these TLSs in junction array fluxoniums.

The high TLS density in frequency and their frequency reconfigurations over time, that the authors extract, makes these TLSs a highly relevant loss-mechanism for the qubit, which means that strategies have to be found to mitigate the detrimental influence of these TLSs to further improve fluxonium qubits.

Strong arguments are provided (coupling strength and density in frequency) that these TLSs are of dielectric nature. Note, that in a recent experiment (arXiv:2501.03661) a magnetic origin could be ruled out.

Except for the conclusion, which is spot on and highlights the importance of the findings for the community, the rest of manuscript seems to be written in a rush. Important information is either absent, fragmented across different sections of the main text, methods and supplement, or must be inferred from an earlier work.

Unfortunately, I have to say that in the current form the manuscript does not meet the quality required for a high-profile scientific publication. Nevertheless, on the basis of the experimental results I highly recommend reconsidering a revised version of the manuscript for publication in Nature Communications.

Below I list 3 major issues I strongly recommend addressing followed by a list of detailed comments and recommendations

1. It almost seems as if deliberate effort went into hiding the incredibly slow readout and reset of the qubit (readout $15\mu\text{s}$ compared to typically $\sim (0.1-1)\mu\text{s}$, reset $35\mu\text{s}$). The fact that the reset is achieved with a highly demolishing readout pulses is only mentioned in the methods section. The unusual readout pulse length is only mentioned in the supplement. No demolition errors are quantified. From the main text the reader can only speculate that the readout is not single-shot, which is highly confusing when the protocols show qubit readout and reset. Given that the readout plays a central role in the protocol and the analysis, I recommend adding the following information in the main text:
 - The readout and reset length and photon number, the dispersive shift and linewidth of the resonator. Besides giving a better understanding of the system, it proves for example, whether the cross-relaxation is indeed inactive during the readout and reset, as the authors claim.
 - The modeling of the protocol as well as the populations are currently based on the assumption that the readout is free from demolition effects. These effects are crucial for the achievable TLS populations and estimation of the TLS lifetimes.

Thus, there should be a number in the main text that quantifies the demolishing errors of the readout (with details in the appendix). The authors may use a similar protocol as in Ref. 20.

- Similarly, I am expecting a plot in the appendix that shows that the reset is more or less independent of the TLS polarization. This plot can be compiled from the current data. That the readout is not free from memory effects is mentioned in the supplement. These state preparation errors of the autonomous reset can drastically influence sensitive parameters like Γ_t . I also want to note that since the reset is most likely not pure, i.e. the qubit population fluctuates from reset to reset and gives on average a $\sim 70\%$ fidelity, there will be deviations from the average dynamics, since the qubit and TLS populations are correlated. To make this point clearer: in the case of active resets, a faulty active reset has a strong influence on the dynamics that limits the polarizability more than proportionally.
2. In my opinion the main finding of the manuscript consists in the observation of long-lived two-level systems in standard aluminum qubits, not only in granular aluminum as first reported. I recommend, positioning the article in this context of the current literature. In contrast, the authors currently focus the article on the “introduction” of relaxation protocols, which are not necessarily faster and exist already in Ref. 20 together with a more profound analysis. The authors should work out more precisely the difference of their protocols and also compare their experimental findings with the results in Ref. 20.
- The discussion whether or not the dynamics is non-Markovian is quite subtle. In the field of quantum master equations, rate equations like the Solomon equations are considered as Markovian. In fact, the Solomon equations are based on the Markov approximation (the dynamics is CP-divisible, i.e. the qubit dynamics is governed by a time-local, but time-dependent master equation). Instead, the Born approximation is not fulfilled (Ref. 25). I recommend speaking of multi-exponential relaxation and the memory: e.g. in the abstract: under Born-Markovian assumptions and could mask such non-exponential
I would also suggest a new title in the following direction: Relaxation spectroscopy of long-lived TLS in Junction-Array Fluxoniums.
(In my opinion, it is quite unfortunate that non-Markovian made it to the title of Ref. 22. If the authors wish to adhere to the non-Markovian nomenclature, I would recommend clarifying what is meant by it.)
 - The different protocols that are “introduced” are essentially a consequence of the challenging readout. The slowly varying qubit transition times as a function of time T , on which the paper is based on, can be seen in Fig. 2c of Ref. 20. Since I do not know of any other device in the community that is read-out that slowly, I highly doubt these specific protocols are relevant for the community. With a standard readout (single-shot and $>90\%$ fidelity) one would rather want to reset the qubit actively with a constant repetition time (as is done in Ref. 20). In this way, one can correct for SPAM errors and demolishing effects (Ref. 20). The only addition in the protocol presented here compared to Ref. 20 is that cooling and heating sequences are alternated instead of looking at the qubit relaxation in between, as in Ref 20. The total measurement time is not necessarily faster. In contrast to the protocol in Ref. 20, following the protocol proposed in this manuscript, one loses the cleaner relaxation data, which is free from state preparation errors, as well as the important information of the “heating + qubit

relaxation from ground state” and “cooling + qubit relaxation from excited state” data. In both protocols, the data throughput is the same and essentially depends on the measurement repetition time.

Instead, I think the only true advantage in this manuscript comes from alternating the polarization, which confines the dynamics more towards the most resonant TLSs. The paragraph ll. 126-133 is thus a bit misleading: 1. data *is* acquired during the polarization sequence to compute the transition rates. 2. the length of the polarization sequence must in any case be adapted to the millisecond time-scales Γ_{qt} of interest. In the present manuscript the authors adapt the protocol on the fly and remeasure.

In short, I recommend to differentiate the protocols more precisely, taking into account the arguments above and directing the focus more towards the experimental findings.

- Throughout the paper the long-lived TLSs are implicitly assumed to be “conventional” dielectric TLSs. ll. 694ff. “Therefore, these results show that low frequency TLS generally have long lifetimes in the phononic environment of planar qubits as well”.

In my opinion, statements like this are too much of an oversimplification, given the fact that the microscopic origin of dielectric TLSs is still mysterious with likely multiple coexisting TLS classes (review Ref. 21 cited by the authors). The fact that the life-time is several orders of magnitude higher than what is usually observed is a strong conflict that can hardly only be explained by the frequency. Note that these long-lived TLSs are still observed at a few GHz in granular aluminum. Just to put these lifetimes into perspective, the phononic-bandgap engineering of Ref. 22 could tremendously enhance the TLS lifetimes from hundreds of ns to only tens of μ s, with three exceptions on a \sim GHz span with qubit relaxation in the ms-range. Another exotic TLS with a long lifetime was reported by two of the authors with a transition frequency of 2.9 GHz. These findings suggest that we should not lump together all these TLSs. The authors may for example use the opportunity to revive the ideas of Schechter & Stamp suggesting two TLS classes (mentioned also in Ref. 21), one at lower frequencies with long lifetimes due to a vanishing coupling to phonons, while the other at higher frequencies has much shorter lifetimes. Regarding the proposed comparison to previous findings, using the measured TLS coherence time of \sim 1.5MHz yields a similar coupling strength $g = 2\pi$ 24kHz for the TLSs in granular aluminum but a higher density of \sim 1 TLS per MHz, which might be expected for a disordered material.

3. I think that the transition rates are quite precisely determined – it follows that there are TLSs (Γ_1) that can be polarized ($\Gamma_{\delta\delta}$). There are TLSs that are long-lived which we know from measurements as shown in Fig. 2. Everything else in the analysis remains vague and does not add much to these findings. Why e.g. does the polarizability spectrum scatter so much and is essentially constant? This plot should reflect the lifetime of the TLSs.

In order to extract the parameters of this convoluted protocols more precisely I suggest that the authors simulate the entire sequence (the simulation is very fast and was already implemented by the authors). For the rather low fidelity qubit reset, one has to decide for a probability distribution. Here one could use the extreme cases of a mixture of 0 and 1 or always use the average reset value. In this way one can cross-check the current analysis, investigate to which extent demolition effects can be neglected and determine errors for the rates, in particular desired for the Γ_t rates.

Detailed comments / recommendations:

- In Fig. 1 the rates $\Gamma_{qt,s}$ should be removed:
1. they are not mentioned in the caption, 2. if these rates denote the surface TLSs, then they are already included in Γ_q , and 3. the figure looks cleaner.
- The paper is based on the fact that the qubit has a long lifetime Γ_q . Please state already in ll. 85-89 the qubit lifetime in the absence of long-lived two-level systems. Otherwise Fig. 2 can easily be misinterpreted with Γ_{qt} being Γ_q . To get a better feeling for the device it would be good to also state the Ramsey T_2 . Can the authors be sure the echo it is not affected by the TLS memory? The reader has to find this information in the caption of Table 1.
- Fig. 2 caption title: "... a bath preparation sequence."
b) ... sum of two exponential decays ...
The two rates Γ_{qt} and $(\Gamma_q + \Gamma_t)/2$ appear from nowhere and are partially wrong, it should be $2\Gamma_{qt}$. You may direct the reader to the analytical formulas in Ref. 25. The rates hold true in the limit $\Gamma_{qt} > |\Gamma_q - \Gamma_t|$.
Fig. 2c): this is not the right example to illustrate a non-polarizable spot. It rather seems that there is a rather strong Γ_{qt} as well as larger Γ_t rate, since the slow rate drops to 1.4ms. Instead, one would expect a low Γ_{qt} to lead to a nearly single exponential decay that exceeds the 2ms of b).
- The choice of using Z and p is not very wise and not consistently used throughout the paper. In the appendix p is also used to indicate qubit populations.
I do not see any benefit of using polarizations. I would suggest to stick to the notations that were used in Refs. 20, 22 and 25. This simplifies the comparison, in particular since many formulas are rederived here. It does not hurt to add further citations and would help the reader to find the original derivations, especially in the appendix.
- Fig. 3: The rather large t_1, \dots, t_4 values are somewhat in conflict with $1/\Gamma_{qt} \sim 20\mu\text{s}$ of Fig. 2. As the authors also explain in the main text, the 4-point exponential fit only makes sense when the bath polarization changes slowly (many TLSs), otherwise t_2 measures a drastically altered TLS population from the previous polarization at t_1 . However, why should we use this sequence in the first place and not always the CD-8 that the authors also mention, especially since the authors do often resolve individual TLSs? The CD-8 gives the same data throughput and is less prone to exponential fit misinterpretations.
My suggestion would be to simplify Fig. 3. In a), explain the constant repetition time sequence CD-8 of the appendix and then show c) and d). Explain in the appendix the version with the different timings and why these protocols are also needed/beneficial. Please use not too many different notations for the same thing, I would suggest that you stick to the transition rates $\Gamma_{up,down}(0)$. It would be very helpful to show the estimated TLS population versus time T.
- Section C beginning: The inverse of T_1 is commonly Γ_1 , there is no need to define something new. The authors also use Γ_2 . The time-dependence (T) of the rates is missing. It is unclear e.g. in Fig.4a) at which time the rates are taken.
In order to judge whether a comparison of the Γ_1 rates makes sense to exclude a bosonic environment the reader must know the TLS polarization that is reached. In Fig. 4a) the points scatter several sigma from the red line, which means in principle

that in some places there is a bosonic environment, however, since the points scatter in both directions, it actually means that the analysis of the data is prone to errors or that the statistical errors are not calculated correctly. The authors simply show that the rates are correlated, which is not at all surprising. Instead, one has to show at each flux position that the rates are identical, for this the authors may use the rates as a function of the time T . In the current form, all statements derived from this plot are not supported.

- The rate Γ_δ and $\Gamma_{\delta\delta}$ are time-dependent. It is not clear at which time they are evaluated. It is not explained how the $\langle Z \rangle$ average is computed exactly.
- ll. 214 formula holds true only for 1 TLS.
- Eq. 4: Z needs the index k .
- It is not explained how Γ_t is extracted. In the methods section the parameter τ_e is introduced (valid for 1 TLS). τ_e should approximately be the inverse of $\eta\Gamma_{qt} + \Gamma_t$. Especially for such a delicate parameter like Γ_t it would be important to get a range of uncertainty.
- ll.71 (on the order of ms^{-1}) What does it refer to? Coupling strength? Qubit loss?
- Subscript labels should not be in italic. In the supplement it is sometimes done correctly.
- For consistency: $\Gamma_{2,k}$ with the comma.
- ll. 31 Grünhaupt et al. Nature Materials 2019 would be a better fit.
- Eq. S6 needs a citation. How is the total dephasing rate defined?
- Eq. S30, the matrix upside down.
- Fig. 1 needs scale bars.
- Fig. S4 colorbar label is missing
- I think at this point it is clear that the manuscript has to be thoroughly revised, such that the overall quality and transparency is improved.